



Natural Hazards
and Earth System
# Extreme wave events in Ireland: 2012 - 2016

Laura O'Brien[1], Emiliano Renzi[2], John M. Dudley[3], Colm Clancy[1], and Frédéric Dias[1,4]

[1]School of Mathematics and Statistics, University College Dublin, MaREI Centre, Dublin, Ireland
[2]Centre for Data Science, Loughborough University, UK
[3]Institut FEMTO-ST, UMR 6174 CNRS-Université de Franche-Comté, Besançon, France
[4]Centre of Mathematics and Their Applications (CMLA), ENS Paris-Saclay, CNRS, Université Paris-Saclay, 94235 Cachan.

*Correspondence to:* Frédéric Dias
(frederic.dias@ucd.ie)

**Abstract.** This paper aims to extend and update the survey of extreme wave events in Ireland that was previously carried out by O'Brien et al. (2013). The original catalogue highlighted the frequency of such events dating back as far as the turn of the last ice age through to 2012. Ireland's marine territory extends far beyond its coastline and is one of the largest seabed territories in Europe. It is therefore not surprising that extreme waves have continued to occur regularly since 2012, particularly

considering the severity of weather during the winters of 2013-14 and 2015-16. In addition, a large number of storm surges have been identified since the publication of the original catalogue. This paper updates the O'Brien et al. (2013) catalogue to include events up to the end of 2016. Storm surges are included as a new category and events are categorised into long waves (tsunamis and storm surges) and short waves (storm and rogue waves). New results prior to 2012 are also included and some of the events previously documented are reclassified. Important questions regarding public safety, services and the influence of

climate change are also highlighted.

## 1 Introduction

The study of extreme wave events in the ocean has become a popular area of research in recent years. Aside from sea-farers, extreme waves impact coastal communities and are of great interest to wave energy companies. This is particularly prevalent in the face of coastal erosion, rising sea levels and uncertainty in how the wave climate will change in a warming world. However,

much of the current research is based on modelling and experiments. Aside from Kharif and Pelinovsky (2003); Tinti et al. (2004); Nikolkina and Didenkulova (2011) and O'Brien et al. (2013), there are few studies documenting the observations of such events.

     The purpose of this paper is to extend and update the the work of O'Brien et al. (2013), a catalogue of extreme waves around the island of Ireland. Ireland's marine territory extends far out into the Atlantic and covers approximately $880,000$

$km^2$. O'Brien et al. (2013) documented extreme wave events extending as far back as 14 680 BP through to 2012, including storm waves, rogue waves and tsunamis. Since its publication, an Irish Costal Protection Strategy Study (The OPW and RPS, 2013) has highlighted the huge number of storm surges that have occurred in Ireland. We therefore incorporate storm surges as an additional category in this paper. We also distinguish between two wave groups, long and short, since their characteristics





are distinct. Our study includes new events that have been identified prior to 2012, recategorises events from the previous catalogue (O'Brien et al., 2013) and extends this catalogue out to 2016.

The paper is organised as follows: Section 2 gives an overview of the categories of ocean waves included in this study, Section 3 revisits the events from O'Brien et al. (2013). New events identified prior to 2012 are outlined in Section 4, and

a catalogue of events from 2012 to 2016 is laid out in Section 5. A wider selection of issues relating to extreme waves are discussed in Sections 6-9: these are boulder deposits, climate change, public awareness and services, respectively.

## 2   Categories of ocean waves

In O'Brien et al. (2013) waves were broken into three categories: storm waves, rogue waves and tsunamis. Storm surges and meteo-tsunamis were included as a subsection of tsunamis as they are both long-period waves with a meteorological origin.

Since this publication, new data (The OPW and RPS, 2013) highlighted a huge number of storm surge events between 1961 and 2005. With this in mind, upon review of the events documented in O'Brien et al. (2013), some have been recategorised as storm surge events. In this paper, we give storm surges its own category due to the large quantity of events that have come to light.

In addition, we feel it is appropriate to divide the wave categories into two streams: long waves and short waves. Generally,

the ratio of depth to wavelength is used as a parameter to differentiate between long and short waves. Long waves can be modelled using a simplified set of equations, called the shallow water equations. Tsunamis and storm surges are typically very long waves, so they are considered shallow water waves. Storm waves and rogue waves are much shorter in wavelength relative to the ocean depth. The dimensionless wave number is $kh$ where $k = 2\pi/\lambda$, $\lambda$ is the wavelength, and $h$ is the water depth. Tsunamis and storm surges have $kh$ values of the order $10^{-3} - 10^{-1}$, while for storm waves and rogue waves $kh$ values

are of the order $1 - 10^3$. Typical values for wavelength ($\lambda$), water depth ($h$), and period ($T$) for storm waves, rogue waves and tsunamis are given in Table 1. The dispersion relation for short waves ($\omega = \sqrt{gk}$) and for longer waves ($\omega = \sqrt{gk \tanh(kh)}$) is used to calculate the range of periods, $T$ in this table. For long waves we calculate the period based only on the upper limit of $\lambda$.

**Table 1.** Classification of short waves (storm waves, rogue waves) and long waves (tsunamis) according to their wavelength, depth and period. Note that storm surges are not included as the range of parameters is difficult to evaluate.

|  | $\lambda$ | $h$ | $T$ |
|---|---|---|---|
| Storm Wave | 10 - 500 m | 5 m - 4 km | 2.5 - 20 s |
| Rogue Wave | 10 - 500 m | 5 m - 4 km | 2.5 - 20 s |
| Tsunami Wave | 1 - 200 km | 50 m - 4 km | 15 min - 2.5 hr |

Note that storm surges are not included as the range of parameters is difficult to evaluate. For instance, it can be difficult to

put a lower bound on the period of long waves. Typhoon Haiyan storm surge lastet less than one hour. However, Roeber and



Bricker (2015) replicated this event showing that it was mainly due to the abrupt breaking of energetic storm waves over the steep reef face rather than a storm surge. Table 1 can be used as a guide, however there are always going to be events that are exceptions to the rule. For example, although $kh$ is generally large for short waves, a rogue wave was measured by a Waverider buoy at Killard (Figure 11) in 39 m depth with a corresponding $kh = 0.74$.

## 2.1 Short waves

### 2.1.1 Storm waves

Storm waves are wind surface waves that reach unusually large amplitude due to forcing by strong winds. For example, storm to hurricane force winds ranging from 10 to 12 on the Beaufort scale have probable maximum wave heights from 12.5 m to 16 m and beyond (The Met Office).

The prevailing wind direction in Ireland is from the south and the west, and on average there can be more than 50 days with gales (10 m wind speeds $> 17.2$ m/s) a year at northern coastal locations such as Malin Head (Met Éireann, 2016c). Figure 1 (a) shows the percentage frequency of wind direction at numerous locations around Ireland, and the percentage of calm days ($< 0.3$ m/s) is circled. Note that while an inland area like Birr has approximately 1 in 10 calm days, coastal areas like Malin Head and Rosslare have approximately 1 in 100 and 1 in 200 days that are calm, respectively. Figure 1 (b) shows the maximum gust speed likely to be exceeded once in 50 years. Note that hurricane winds are those that exceed 32.7 m/s and the contours on this map range from 44 m/s to 50 m/s .

In addition, waves approaching a cliff can be significantly amplified by a variable coastal bathymetry. Herterich and Dias (2017) have shown that the waves could be amplified nearly 12 times around the Aran Islands (3 islands off Co. Galway, on the west coast of Ireland).

With strong winds so prevalent in Ireland and coastal bathymetries potentially favouring significant wave amplification, storm waves are an important category of extreme ocean events in this catalogue.

### 2.1.2 Rogue waves

Rogue waves are large-amplitude waves surprisingly appearing on the sea surface (Kharif and Pelinovsky, 2003). They seem to appear from nowhere with a height $2 - 3$ times that of the surrounding sea state, exist for a short time and then disappear. The mathematical criteria

$$H/H_s > 2 \ \text{ and/or } \ \eta_c/H_s > 1.25$$

where $H$ is the (trough-to-crest) wave height, $\eta_c$ is the crest height and $H_s$ is the significant wave height, are commonly used (Dysthe et al., 2008). Rogue waves are also referred to as freak waves, monster waves or king waves. Once thought to be folklore of seafarers, they are now accepted as an important class of wave. This is due to recent scientific investigations motivated by account of huge waves hitting ships (Kharif and Pelinovsky, 2003; Didenkulova et al., 2006) and measurements of unusually large waves from oil-platforms (Olagnon and Prevosto, 2004; Magnusson and Donelan, 2013; Christou and Ewans,





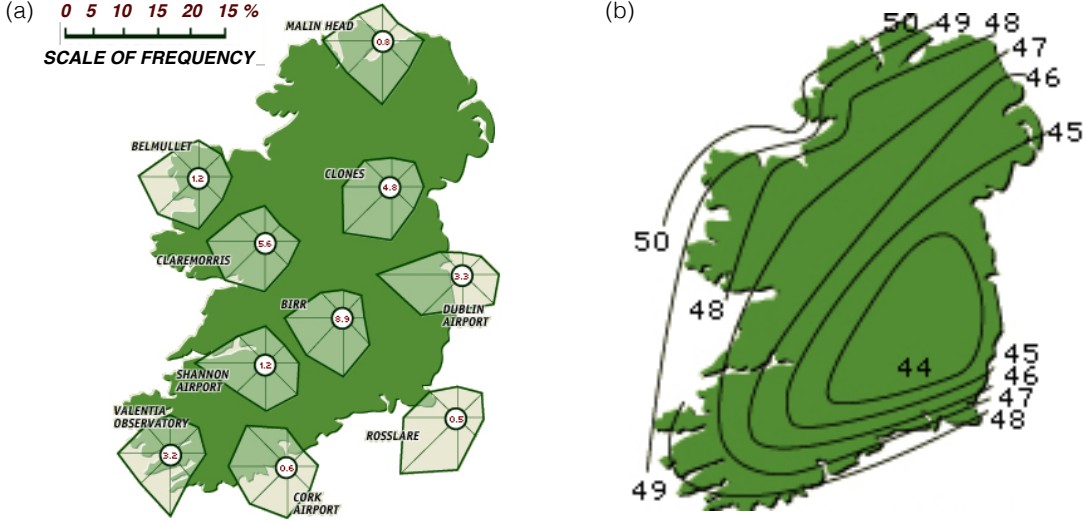

**Figure 1.** (a) Percentage Frequency of wind direction with circled number representing the percentage of calm days. (b) Maximum gust speed likely to be exceeded once in 50 years (m/s). (Met Éireann, 2016c)

2014; Fedele et al., 2016; Cavaleri et al., 2016; Donelan and Magnusson, 2017). Rogue waves are random in nature, occur in both deep and shallow waters, and may act as a single wave or a group of waves. It is almost certainly the case that different mechanisms can contribute to the formation of rogue waves, including both linear and nonlinear processes influencing waves both in the local vicinity of a wind sea, as well as propagating swell. For example, linear superposition and directional focussing

effects could readily increase local wave height in the presence of local wind, whilst nonlinear instabilities could increase local wave group amplitude for a propagating swell.

Work using analogies between ocean deep water waves and light propagation has indeed been able to confirm many predictions (Peregrine, 1983) of nonlinear deep water wave growth and decay in an optical environment (Kibler et al., 2010, 2012). However, Fedele et al. (2016) recently investigated real world ocean rogue waves and showed that their observed results could

be explained by directional focussing. A discussion of linear and nonlinear rogue wave mechanisms is given in Dudley et al. (2013).

Draper (1971) discusses the expected frequency of large waves in the context of the statistical distribution of the maxima of a random function theory by Cartwright and Longuet-Higgins (1956). In particular, he points out that the theory indicates we should expect one in 1175 waves to exceed three times the average wave height ($\sim 1.9H_s$) and one in 300,000 waves to exceed

four times the average wave height ($\sim 2.5H_s$).

Note that field measurements of maximum wave heights ($H$) in constant depth generally do not exceed $0.55h$ (where $h$ is water depth) and tend to break before reaching this height (Nelson, 1994; Babanin et al., 2001). Experimental results by Grue et al. (2014) closely agree with this for strongly breaking waves ($H < 0.56h$) but exceed this value for moderately breaking



waves ($H < 0.63h$), while numerical results (Zhang and Schäffer, 2007; Yang et al., 2013) again exceed this value but not by much ($H < 0.6h$). Therefore, we expect $H = 0.55h$ to be a good indicator of the maximum rogue wave heights in the ocean. While $H < 0.63h$ would be the upper limit in idealised situations.

## 2.2 Long Waves

### 2.2.1 Tsunamis

Tsunami waves are mainly generated by earthquakes, landslides or volcanic activity displacing large volumes of water. They can have wavelengths of the order of 100 km and travel at around 800 km/hr. Although waves are usually only a few tens of centimetres high in the open ocean they can become catastrophically big as they are compressed and slow down along the shoreline. They have devastated countries at severe human cost, wiping out miles of coastline, towns and villages on a path of destruction. The recent events in Japan in 2011 and Indonesia in 2004 are a stark reminder of their destructive power.

It is common that the first sign of a tsunami is an extreme withdrawal of the sea followed by a wave that seems small in the distance, but which grows rapidly and which can be followed by successive waves sometime after. The first wave is not always the largest (Stefanakis et al., 2011; Okal and Synolakis, 2016). Without the proper infrastructure and education of the warning signs and evacuation methods, tsunamis can have deadly effects on coastal communities.

Small tsunamis can also occur. They appear as disturbances with the same generation mechanisms and characteristics of a tsunami without being life threatening. There are cases in harbours where the tide has unexpectedly risen and fallen repeatedly every few minutes over the course of an hour or two. For example, in Rabaul, Papua New Guinea a small tsunami occurred on the 29 March 2015, "residents noticed the sea level rose slightly, prompting ocean water to flood the parking lot of a shopping center near the beach" (National Oceanic and Atmospheric Administration, 2016).

Meteo-tsunamis are waves with tsunami like characteristics but are caused by air pressure disturbances often associated with fast moving air such as squall lines. Their development depends on the characteristics of the disturbance (speed, intensity, direction) and they can be magnified by resonances associated with the depth of the water or when travelling in semi-enclosed water bodies (National Tsunami Hazard Mitigation Program, 2015). Although there are not many documented incidents of meteo-tsunamis relative to regular tsunamis, and even less of meteo-tsunamis that have become dangerous, a 2 m wave in 2013 injured three people in New Jersey and a 5 m wave killed three people in Nagasaki Bay, Japan in 1979 (Kathleen Bailey and Welty, 2014).

If there is no known origin of a tsunami like wave it can be difficult to differentiate between a small tsunami and a meteo-tsunami. For example, underwater landslides can occur without notice other than the generation of a wave, so it is important to consider different generation mechanisms in each case.

### 2.2.2 Storm surges

A storm surge is an unexpected rise in sea water level generated by a storm. The low-pressure area near the storm's eye reduces the weight of the air over the ocean. This creates a swell in the sea which is pushed towards the coast by the strong winds. As





the storm approaches the coast, the combined effect of the low pressure and the violent winds makes the water pile up along the shore. It is a long-period wave and can have particularly destructive effects in coastal areas where there is a significant difference between low tide and high tide.

In Europe, storm surges alone usually do not generate coastal flooding. Coastal flooding occurs when a storm surge hits the shore together with high tides. To the coastal communities, the effect of the surge-tide coupling is the same as a small tsunami, despite the physics being very different.

The climate in Ireland is dominated by the Atlantic Ocean, in particular the structure of the polar front over the Atlantic and the associated weather systems that travel across Ireland. These systems can often amplify and become large scale depressions that move north eastwards across the North Atlantic and pass to the northwest of Ireland (Met Éireann, 2016b). Sometimes this can lead to storms travelling across the ocean that initiate storm surges and lead to coastal flooding.

The period of storm surges can range from 2.5 hours to a day and excludes wind waves and choppy sea, since they are characterized by periods smaller than a minute. The only formal difference between a storm surge and a meteo-tsunami consists in the difference between their maximum periods. The maximum period for a tsunami does not exceed several hours, while storm surges may last up to a day.

## 3   Events 14 680 BP - 2012 Revisited

The catalogue of events from O'Brien et al. (2013) has been revisited here. In some cases events are reclassified as storm surges based on descriptions of the event. In other cases, new references have come to light providing additional information about the event. Table 2 lists every event from O'Brien et al. (2013) and indicates whether the event has been upgraded to a storm surge or if new information has been found. Section 3.1 elaborates on each of these events.

### 3.1   Additional Information

#### 3.1.1   1894 The Mullet Peninsula, Co. Mayo [S1]

The 29 December 1894 event has been upgraded to a storm surge due to the eye witness description of a terrible gale that was blowing all night, with "Green seas... going over our houses", and all the rooms "filled with the sea" (Ryan, 1895). On reflection this account is more akin to the description of a storm-induced coastal flooding event rather than an isolated storm wave.

#### 3.1.2   1881: Calf Rock, Co. Cork [S2]

A new reference to this event gives some more information about what took place on the rock. The description of flooding suggests that this was actually a storm surge event. "It was blowing a stormy gale from N.W., accompanied by lurid flames of lightning every minute... three heavy waves struck the tower... The three of us were in the basement story all this time, and when the wave struck the tower, breaking it off above the basement, the rush of water, bricks, and all came down into the





**Table 2.** List of events from O'Brien et al. (2013) with any new information or revisions.

| Label | Event | Update |
|---|---|---|
| S1 | 1837, 1861, 1894, 1935, 1987, 1988 & 1989: The Mullet Peninsula, Co. Mayo | 1894 recategorised as storm surge |
| S2 | 1869 & 1881: Calf Rock, Co. Cork | New reference for 1881 event, recategorised to storm surge. |
| S3 | 1839: Night of the Big Wind | Likely that this includes a storm surge at the Cliffs of Moher |
| S4 | 1864: Valentia, Co. Kerry | |
| S5 | 1877: Railway Lines, Co. Dublin & Co. Wicklow | Recategorised as storm surge |
| S6 | 1899: Greenore, Carlingford Lough, Co. Louth | |
| S7 | 1941: Inisheer Lighthouse, the Aran Islands | Recategorised as storm surge |
| S8 | 1945: Rosslare, Co. Wexford | Recategorised as storm surge |
| S9 | 1951: Kilkee, Co. Clare | |
| S10 | 1953: The Aran Islands | Recategorised as storm surge |
| S11 | 1962: Co. Cork | |
| S12 | 1974: Kilmore, Co. Wexford | |
| S13 | 1979: Fastnet Race | |
| S14 | 1982: Ventry, Co. Kerry | |
| S15 | 1985: Fastnet Rock Lighthouse | |
| T1 | 14,680 BP: The Barra Fan, Peach Slide | |
| T2 | 8200 BP: Storegga slide | |
| T3 | 1755, 1761, 1941 and 1975: The Lisbon, Portugal Tsunamis | Information about an additional tsunami in 1969, and tide gauge recordings from 1969 and 1975. |
| T4 | 1767: The River Liffey, Dublin | |
| T5 | 1841: Kilmore, Co. Wexford | New reference |
| T6 | 1854: Kilmore, Co. Wexford | New map |
| T7 | 1894: Galway Bay & The Atlantic (Festina Lente & Manhattan off map) | |
| T8 | 1922: Ballycotton, Co. Cork | |
| T9 | 1909: Westport Quay, Co. Mayo | Suspected meteo-tsunami or tsunami due to landslide |
| T10 | 1910: Cork, Waterford, Southampton, Jersey, Dublin & Ilfracombe | |
| T11 | 1912: Bray, Co. Wicklow | |
| T12 | 1932: Inishowen, Co. Donegal | Recategorised as storm surge |
| R1 | 1852: Inis Mór, The Aran Islands | |
| R2 | 1883: Youghal , Co. Cork | |
| R3 | 1899: Kilkee, Co. Clare | |
| R4 | 1914: Iniskeeragh, off Donegal | |
| R5 | 1936: Dundalk, Co. Louth | |
| R6 | 1972: Mullaghderg, Donegal | |
| R7 | 2004: LE Róisín, off Donegal coast | |
| R8 | 2006: off Portrush, Co. Antrim | |
| R9 | 2006: Ardglass, Co. Down | |
| R10 | 2007: Doonbeg, Co. Clare | |
| R11 | 2007: Valentia Island, Co. Kerry | |
| R12 | 2011: Swanland, off Bardsey Island, The Irish Sea | |
| R13 | 2011: Largest wave recorded in Ireland | This record has since been exceeded (see Section 6) |

See additional information in Section 3.1.





basement story, washed us down to the door, and we were up to the armpits in water.... when the sea came in we had to jump on the chairs." (*The Horsham Times, VIC, Australia*, 28 January 1882, State Library Victoria).

### 3.1.3 1839 The Night of the Big Wind [S3]

The 6-7 January 1839 was one of the most damaging storms Ireland has ever encountered. This was originally listed under the storm waves category in O'Brien et al. (2013). However upon reflection of some of the descriptions it is likely that a storm surge occured alongside storm waves in some regions. The description of how high the waves were at the Cliffs of Moher and the Aran Islands indicates that a storm surge most likely occured, "[T]he waves actually broke over the tops of the Cliffs of Moher... the ocean tossed huge boulders onto the cliff tops of the Aran Islands" (Bunbury, 2005).

### 3.1.4 1877 Co. Dublin - Co. Wicklow [S5]

This event in January 1877 was most likely a storm surge going by the description of flooding on the train line, "The train line between Kingstown (Dún Laoghaire) and Bray was considerably damaged by recent floods" (*The Irish Times* 25 January 1877).

### 3.1.5 1941 Inisheer Lighthouse, Aran Islands, Galway Bay [S7]

The reference to severe flooding at Inisheer lighthouse in January 1941 suggests that this event was a storm surge (Williams and Hall, 2004).

### 3.1.6 1945 Rosslare, Co. Wexford [S8]

On the 18 December 1945 gales and high tides hit many coastal towns. The description of an "exceptionally high tide" that washed away part of the cliffs in Rosslare bay indicates that this event was likely to have been a storm surge (*The Irish Times* 19 December 1945).

### 3.1.7 1953 Aran Islands and Irish Sea [S10]

During a storm in 1953 a large number of megaclasts at Gort Na gCapall on the Aran Islands were shifted (Williams and Hall, 2004) indicating that it was likely a storm surge had occurred.

### 3.1.8 1969 and 1975 Lisbon Tsunamis [T3]

In addition to the 1755, 1761, 1941 and 1975 Lisbon, tsunamis documented in O'Brien et al. (2013), there was another significant earthquake and tsunami near Lisbon on the 28 February 1969. The tide gauge recordings at Belfast harbour show small oscillations which may be the arrival of this tsunami, see Figures 2. In addition, another tide gauge recording has been located for the 1975 tsunami. The recording is taken at Belfast harbour and is shown in Figure 3.


(a) 28th February 1969      (b) 1st March 1969

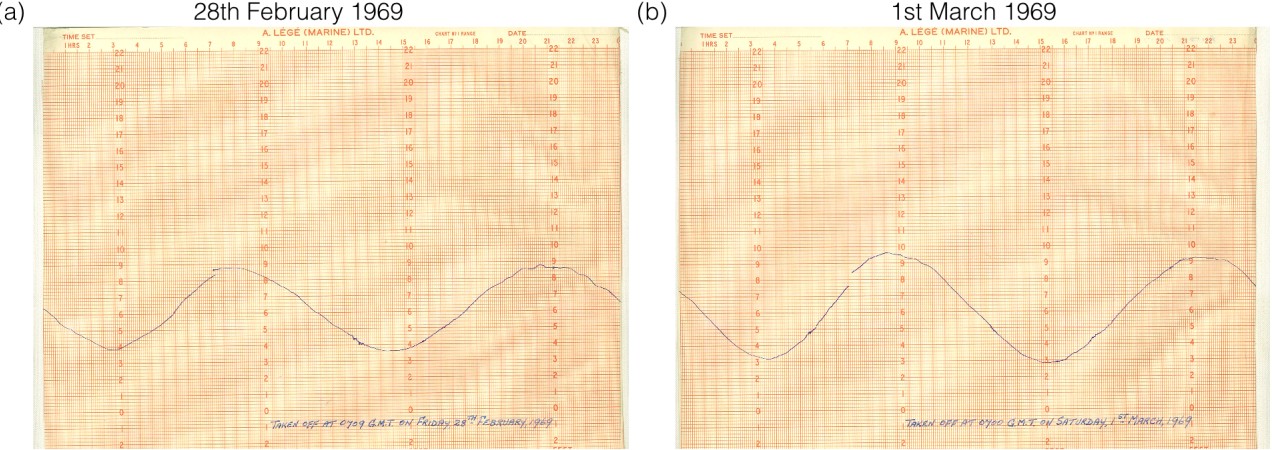

**Figure 2.** Tide gauge recordings at Belfast harbour on (a) 28th February 1969 and (b) 1st March 1969. Minor oscillations can be seen upon close inspection. Retrieved through personal contact with the the Permanent Service for Mean Sea Level (2016), (Holgate et al., 2013).

(a) 23rd - 26th May 1975      (b) 26th - 28th May 1975

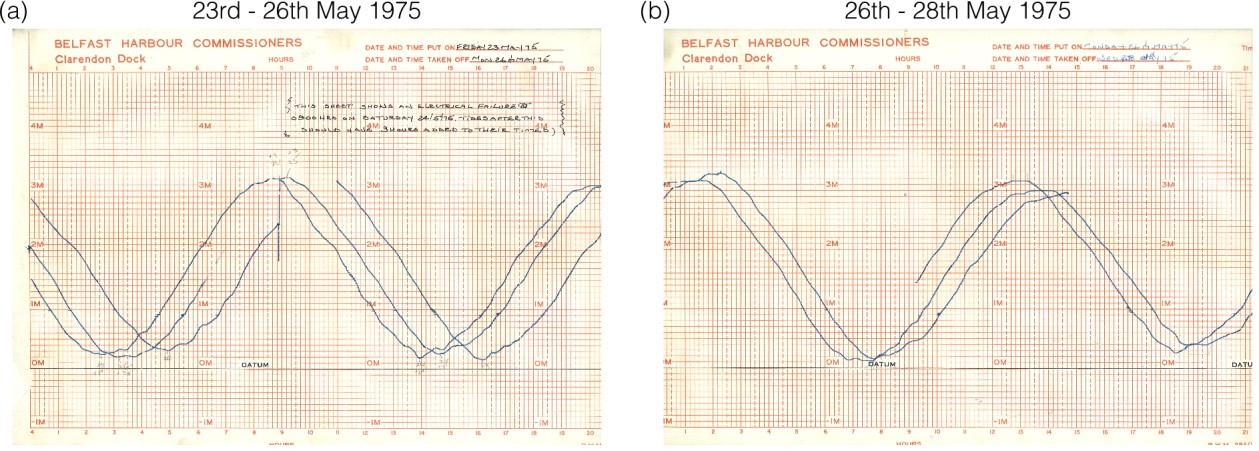

**Figure 3.** Tide gauge recordings at Belfast harbour on (a) 23rd - 26th May 1975 and (b) 26th - 28th May 1975. Minor oscillations can be seen upon close inspection. Retrieved through personal contact with the Permanent Service for Mean Sea Level (2016), (Holgate et al., 2013).




### 3.1.9 1841 Kilmore, Co. Wexford [T5]

O'Brien et al. (2013) only had information about this event from the RAS Historical Tsunami Database which listed it as a questionable tsunami. However, another reference has come to light with a description of what occurred at Kilmore on the 12 September 1841 (Milne, 1844). "the attention of the inhabitants was attracted about noontime, to a number of short, loud, but rather smothered reports, like cannons, and it was supposed that they proceeded from some ship bewildered by the fog. The tide had flown pretty well at the time, and the fishing boats in the pier were all afloat, when, in a space of two or three minutes, the water receded from the pier, and some walked dryshod where the short pace before the boats had been floating in five or six feet of water. In the course of a few minutes the waters began to return, much in the same way as they had receded, and the tide continued to rise for the usual time. There was no extraordinary commotion, only an increased surf. After repeated rolls in thunder, and some heavy showers, the sky cleared up. It is the belief generally that this singular motion was the effect of an earthquake, whose shocks have of late been so frequently experienced in Scotland." The recession and return of the tide in such a short period of time would suggest that this was definitely a tsunami, but it is not clear what caused it. The weather preceding the wave is described as "a misty dark day, with the wind SSW to S... the low growel (sic) of distant thunder was heard, and the wind lulled, which rendered the fog more dense". This might suggest a meteorological origin, however this is only speculation.

### 3.1.10 1854 Kilmore , Co. Wexford [T6]

A map depicting the tsunami at Kilmore Quay on the 16 September 1854 has come to light since the publication of O'Brien et al. (2013), see Figure 4.

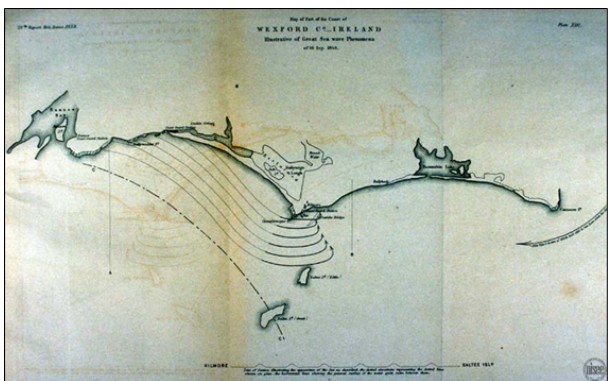

**Figure 4.** A tsunami at Kilmore Quay on 16 September 1854 (Mallet and Mallet, 1854).





### 3.1.11 1909 Westport Quay, Co. Mayo [T9]

O'Brien et al. (2013) likened this event to the 2011 mild tsunami that occurred on the south-west coast of England because of the similar indictions of static in the air. The 2011 event was originally reported as a probable meteo-tsunami. However, since then it was reported that the 2011 event was likely caused by an underwater landslide (*BBC News*, 29 June 2011). Therefore, it is still not clear what caused the 1909 tsunami like wave in Westport, but an underwater landslide or meteorological influences are likely factors.

### 3.1.12 1932 Inishowen, Co. Donegal [T12]

The description of this event indicates that more likely it was a storm surge, not a meteo-tsunami. In particular the "Violent Gales" and tide rising to a height "not seen for very many years" would correspond to a storm surge.

### 3.1.13 2011 largest wave ever recorded in Ireland [R13]

On the 13 December 2011 the M4 weather buoy registered a 20.4 m wave which was the largest wave ever recorded by the buoy network to date, however this record has since been broken.

The Kinsale Energy Gas Platform recorded a maximum wave height of 25 m on the 12th February 2014 during a severe windstorm (Met Éireann, 2016a).

Then in 2016, Storm Jake hit Ireland with severe winds of up to 133 km/hr on 2 March, and a 30.96 m wave was recorded at 00:30 on the wave buoy Belmullet Berth B (Figure 14). This could be interpreted as a suspicious reading given how big it is, however the nearby wave buoy Belmullet Berth A also recorded a 26.35 m wave at 03:30 (Figure 14) indicating that there were very rage waves in the region. Since the buoy network was only set up in November 2000, it is not surprising that the record has been broken since 2011.

In addition, analysis of a wave rider buoy at Killard (off the W coast) identifies a 33.96 m wave on 26 January 2014 at 22:30 (Figure 11). Again, this could be interpreted as a suspicious reading and we don't accept this as the largest wave recorded in Ireland. However, it is worth noting that the M4 buoy (off the NW coast) recorded a wave of 23.44 m on the afternoon of the same day.

## 4 New Events up to 2012

### 4.1 Storm Waves

#### 4.1.1 1588 Spanish Armada

The Spanish Armada, consisting of 130 ships and 29,450 men were driven off-course by bad weather during their attempt to return home through North Atlantic after their effort to conquer Protestant England. 24 ships were wrecked off the Irish coast from Antrim in north to Kerry in south (O'Hara, 2013).




### 4.1.2   1895 Dun Laoghaire

15 lifeboat men lost their lives on 24 December 1895 when they were called to rescue a Finnish vessel, SS Palme which had run aground in Dublin Bay. This is the worst tragedy to have occurred in the history of the Royal National Lifeboat Institution (RNLI). High winds, rainstorms and flooding occurred across the country throughout December, culminating in ferocious gales

in the final week of the year and 9 ships wrecked on shore or sunk at sea. The SS Palme left Liverpool on 18 December and immediately battled gales attempting to head south for many days without making much headway. On the morning of the 24 December the captain, with an exhausted crew, decided to make a run for Dublin Port. However, as the visibility worsened and they battled constant rain squalls they headed for Kingstown Harbour but were unable to navigate to safety so let their anchors go outside the West Pier. Two lifeboats were sent to rescue the crew and as the first approached the ship it was hit by a huge

wave and capsized. The ship's crew attempted to launch one of their boats to rescue the men but it was smashed against the ship's side in heavy seas. The second lifeboat also capsized twice and lost half her oars but righted and the crew managed to get to safety at Blackrock. Thirteen of the crew's bodies were washed up on Christmas day and the other two several days later. They included a father and son, two pairs of brothers and left a total of thirty-four children fatherless. Astonishingly, all nineteen people aboard the ship were saved when the weather abated on the 26 December. (Louth, 2013).

There is a memorial to the crew at Dun Laoghaire shown in Figure 5.

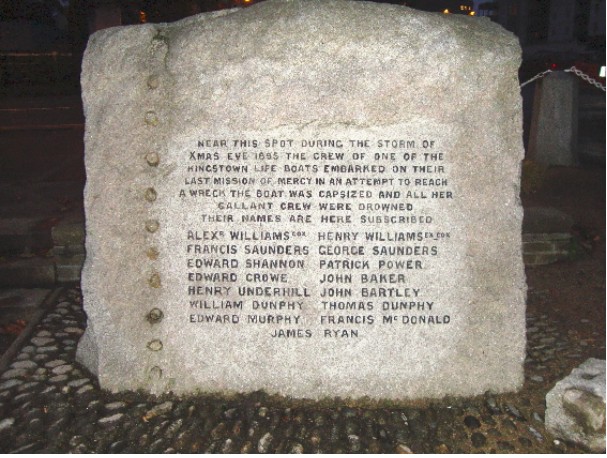

**Figure 5.** Granite memorial on the old lifeboat station wall, Dun Laoghaire Hohenloh (2009).

### 4.1.3   1903 Storms across Ireland

The period ending February 1903 was very stormy with depressions from the Atlantic bringing very unsettled weather across Ireland. The night of 26th to 27th was probably the most severe storm since "The Night of the Big Wind" (Met Éireann, 2016a). On the 26th, the Lady Disdain, a 25 ft yacht was wrecked in Valentia harbour, Co. Kerry, on its journey from Kingstown (Dun




Laoghaire) to Nenagh. Thursday last "there burst on the bay the hurricane which swept the whole of Ireland". At about 4pm "the great wind came on" and "Great waves rode down the harbour tossing the biggest vessels like cockieshells". It was driven ashore and struck rocks. (*The Irish Times*, 5 March 1903) Also associated with this bad stretch of weather was the drowning of a 9 year old boy at Portrush, Co. Antrim, while he was on a patch of strand beneath the promenade. It was windy and the waves "frequently at long intervals" swept over the rocks and sand. The boys cap blew off and when he went to get it a "tremendous sea broke into the inlet, knocking him against the rocks, and on receding carried him out with it". (*The Irish Times*, 28 February 1903)

### 4.1.4 1927 Major Storm off the West Coast of Ireland

On 28 October 1927 a major storm swept across the northern half of Ireland and Britain, and took the lives of 45 people at sea. 9 fishermen drowned at Lacken Bay, Co. Mayo when the storm suddenly blew up and their boats were driven against the rocks. 10 people from Inishkea Islands, 9 people from Inishbofin and 26 fishermen in Cleggan Bay also lost their lives. (Met Éireann, 2016a)

### 4.1.5 1935 Arranmore boat disaster

On 9 November 1935, a small boat carrying 20 people from Burtonport harbour, Co. Donegal to Arranmore Island hit rocks and sank. There was only one survivor and among the casualties were 12 people under 30 years of age, and 7 from the same family. Many were returning from Scotland after picking potatoes for the season. Although the accident reportedly occurred in a heavy swell, the cause was due to the boat hitting rocks. It is therefore unlikely that this can be attributed to a wave. However, the stretch of sea is known for being treacherous, and 15 people died in three other incidents over the same area. (*The Irish Times*, 11 November 1935, and RTÉ (2016c): *50th Anniversary of Arranmore Boat Disaster 1985*)

### 4.1.6 1965 Achill Island

A 37 year old farmer was swept out to sea by huge waves during a storm while he was beach combing with friends. (*Irish Independent* 19th January 1965).

### 4.1.7 1975, 1976 off Rathlin O'Birne Island, Co. Donegal

On 7th January 1975, after leaving Burton Port, the Evelyn Marie fishing trawler was hit by a wave and smashed up against the reef off Rathlin O'Birne island. The boat split in two and all 6 onboard lost their lives. On 23rd November 1976, another fishing trawler, the Carraig Una was lost in very similar circumstances to the Evelyn Marie off Rathlin O'Birne. All 5 crew members died. (RTÉ, 2016a)





### 4.1.8   1981 off Killybegs, Co. Donegal

On 1st November 1981 the fishing trawler, the Skifjord was battered by a sudden storm and a "mighty wave" wedged the boat up against a reef. The sturdy vessel was "smashed open" and the boat sank, 5 of the 9 crew members died. (RTÉ, 2016a)

### 4.1.9   1983 The Doolin Drownings, Co. Clare

On 31st July 1983, the second day of the Lisdoonvarna rock festival, 8 men aged between 17 and 30, including 3 brothers, drowned at Trá Latháin beach. At about 4:30 pm about 12 people went swimming off the Doolin beach and got into trouble. There were warning signs at the beach entrance but it seems that the group had entered the beach from a different area so didn't know of the dangers. The cause of the drownings is not clear but numerous different factors are reported to have contributed to the accident. (*The Irish Times*, 1 August 1983) An RTÉ documentary (RTÉ, 2016b) suggests that a strong coastal current, a

subterranean river, and a turning tide combined to create treacherous conditions. After the tragedy, the superintendent at Doolin Garda (Police) station warned people that the beach had a drastic drop of about 40 ft (12.2 m) which could not be seen even when the tide was partially in. He also mentioned severe undercurrents and shifting sands. (*The Irish Times*, 3 August 1983)

### 4.1.10   1991 Storm

From the 20th December 1990 to 7th January 1991 vigorous depressions were moving eastwards well to the north of Ireland,

bringing strong winds across Ireland (Met Éireann, 2016a). Numerous rescues by the RAF sea King helicopter took place including a rescue of crew members on board a sinking fishing vessel after a wave split her hull (*The Irish Times* 28 December 1990). A tanker also sank in the Irish Sea after being battered by winds (*The Irish Times* 7 January 1991). Rocks, boulders and waves caused huge damage on Arranmore, off Co. Donegal. Large areas of farmland were destroyed by "hundreds of tons of rocks washed ashore by the fury of the seas", an excavator was required to clear huge boulders off Chapel Road, while several

homes were damaged by waves. Harbours at Bunbeg, Magheragallon, Burtonport and Portnoo, were affected by high tides and boats destroyed (*The Irish Times* 7 January 1991). It was reported that waves reached 12 m (40 ft) at the harbour on Tory Island, off Co. Donegal (*The Irish Times* 25 December 1990).

A deep depression on the 5th-6th January 1991 brought strong gale force winds across Ireland. Met Éireann reported that although there were no wave observations, a reliable wave model predicted a significant wave height of $13 - 15$ m in deep

water off the west coast, classified as phenomenal. Given this prediction they said it was reasonable to expect that individual waves reached $25 - 30$ m (Met Éireann, 2016a).

### 4.1.11   1998 Extreme storms

Severe weather hit Ireland and Europe on the 4th January 1998, causing major disruption to ferry and airline services, and the rescue of 10 fishermen from their sinking trawler 190 miles south of Castletownbere, Co. Cork. They were under tow after

their engine failed and when the tow rope broke and they were "at the mercy of the huge seas" in "force 11 gales" ($103 - 117$





km/hr). The rescue helicopter captain said he had "never seen anything like the conditions" with waves up to 70 feet [21 m] high covered by "20 feet [6 m] of mist" because the wind was so strong. (*The Irish Times*, 5 January 1998)

### 4.1.12  1999 Kilkee, Co. Clare

On the 28th December 1999 a young woman was swept into the sea at Kilkee, Co. Clare while out walking with friends. It is unclear whether this event should be categorised as a storm wave or storm surge since it occurred at the same time as there was significant flooding. However, we include it as a storm wave. (*Sunday Independent* 2nd January 2000)

### 4.1.13  2005 Doolin, Co. Clare

3 men died on 31st October 2005 when the car they were sleeping in fell off a cliff into the sea at Doonagore Bay, Co. Clare. Newspaper reports suggest that they may have accidentally knocked the car out of gear while sleeping, however we will never know what actually happened. Gales force winds and a high swell hampered search operations. (*The Irish Times*, 4, 8, 10 November 2005).

### 4.1.14  2006/7 Kilpatrick, Co. Wexford

During the winter of 2006/7, storms caused about "15 metres of coast collapse into the sea" including a section of an access road to local houses. The road has not yet been repaired and coastal erosion in this area continues to occur. While some locals would like protection from coastal erosion put in place, it is a tricky situation as the region is also a Special Area of Conservation, home to sandmartins, a protected species of bird. Coastal erosion continues in the region with more recent storms. (*The Wexford People* 18 August 2010, *The Irish Times* 14 January 2014)

## 4.2  Rogue Waves

### 4.2.1  1962 off Co. Cork

On the 9th March 1962, a Spanish fishing vessel, *Maria Somenque* had the "entire front structure of the wheelhouse torn from its foundation" by a "freak wave". (*The Irish Press 10th March 1962*).

### 4.2.2  1969 off Co. Cork

Draper (1971) shows a record from the 12th January 1969 taken by the Commissioners of Irish Lights with a shipborne wave recorder on *Daunt* light-vessel off Cork showing a wave that is $4.1$ times the average wave height. Given that the average wave height would usually be around $0.63$ times the significant wave height, this wave would be approximately $2.6$ times the significant wave height. See Figure 6.



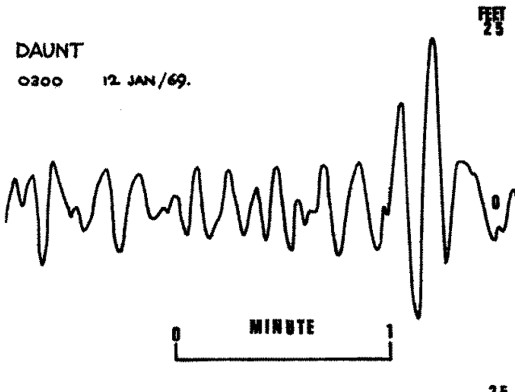

**Figure 6.** Wave recorder trace taken on 12th January 1969 (Figure 1 in Draper (1971))

### 4.2.3 1994 Bridges of Ross, Co. Clare

Professor Paul Wignall from the University of Leeds gave an eye witness account (email communication) of a possible rogue wave event that occurred at the Bridges of Ross, Co. Clare, in March 1994 while on a field trip. "...the wave was 6 feet [1.8 m] higher than the highest part of the Bridge of Ross which would make it approx 60 foot [18 m] higher than typical sea level height. It was quite a stormy/choppy day... People... on the bridge... thought the wave would reach them and started to run across the field behind... the wave actually broke into the gully in front of the ridge. I saw the wave a couple of seconds before it struck and it was a vertical wall of water rather than a typical breaking wave. I've been back several times... and no wave has come close to that one - a pretty memorable event."

### 4.2.4 2006 Tullig Point, Co. Clare

On the 29 October 2006 a man and his friend were swept into the sea by a wave while fishing off rocks near the village of Cross (near Kilkee), Co. Clare. While his friend managed to reach safety within minutes, unfortunately he drowned. (*The Irish Times* 31 October 2006 and 23 December 2006).

### 4.2.5 2006 Blue Pool, Co. Clare

A Latvian man died when he was swept into the sea by a wave while fishing at Blue Pool, between Kilkee and Doonbeg, Co. Clare, on the 5th November 2006. (*The Irish Times* 5 November 2006).

### 4.2.6 2007 Lettermore, Connemara

A man was swept out to sea by a "freak wave" as he was tending livestock near Lettermore, on the Connemara coast, on the 3rd January 2007. (*The Irish Times* 3rd January 2007).



### 4.2.7    2007 Blue Pool, Co. Clare

On the 1st July 2007 a Moldovan man was killed after he was washed from the rocks at Blue Pool, between Kilkee and Doonbeg, Co. Clare, by a "rogue wave". The newspaper report notes that this spot is "notorious for freak waves", citing the 2006 incident also at Blue Pool and the 2006 Tullig Point incident which is nearby. (*Irish Independent* 02 July 2007).

### 4.2.8    2009 West Cork

A man and his son vanished while fishing near Coominches, Co. Cork. on the 12th July 2009. The cause of their disappearance is unknown. The body of the man was recovered. (*BreakingNews.ie* 15 July 2009).

### 4.2.9    2010 Ardfield, Clonakilty, Co. Cork

A man died after fishing by the rocks of a bay near Ardfield, Co. Cork on the 16th August 2010. Superintendent Pat Maher said conditions were misty at the time of the incident and that it was highly likely he had either been swept into the water by a wave or had lost his footing on the rocks while fishing. (*Irish Examiner* 19 August 2010).

### 4.2.10    2011 Ross Bay northern shore of Loop Head Peninsula

In May 2010 a man swept from rocks while fishing at Ross Bay on the northern shore of Loop Head Peninsula. (*Irish Examiner*, 27 August 2011).

## 4.3    Storm Surges

### 4.3.1    1967-2005 Storm surges that caused coastal flooding

Table 3 shows the historic major coastal flooding events across the Republic of Ireland, in the period 1967-2005. A major coastal flooding event is defined here as an inundation in excess of $0.5$ m of an otherwise dry coastal area. Such a level, though usually not harmful to human life, can indeed damage houses and electric grids, interrupt business and transport services, usually causing large financial losses in relief and reconstruction costs. As an example, Insurance Ireland recently announced that the estimated cost of claims relating to storm damage and flooding in December and early January 2014 is approximately $46$ million euros (Insurance Ireland, 2014).

The dataset was extrapolated from the historic storm surge archive published in the Irish Coastal Protection Strategy Study (ICPSS) by the Office for Public Works (The OPW and RPS, 2013). In that study, the Irish coastline was originally divided into six areas, namely North East (NE), South East (SE), South (S), South West (SW), West (W) and North West (NW). For each area, the ICPSS reports a number of storm surge events together with the relevant water level records obtained from local gauges. By cross-referencing the ICPSS archive with technical reports issued by County or City Councils found on the National Flood Hazard Mapping portal (The OPW, 2016), or newspaper articles obtained from the Irish Newspaper Archives, we identified those storm surge events that generated coastal flooding in excess of $0.5$ m.





Note that the table includes events prior to 1961 and after 2005 since the assessment period varies between reports (S: 1967 - 2006, SE and NE: 1959 - 2005, SW, NW and W: 1959 - 2009).

Inland areas can also be affected by storm surges, for instance if an area is connected to the ocean by an estuary or river. Therefore, some inland flooding events have been included as coastal flooding events in this paper. These are restricted to events within ∼ 10 km of the coast. It is also important to note that there are many storm surge events that have reports of major coastal flooding associated with them, however no inundation depths are reported, this is particularly true in earlier years. Therefore, there may be storm surge events that caused coastal flooding in excess of 0.5 m that are not included in this catalogue due to a lack of available information.

Between 1961 - 2006 we identified 40 periods where coastal flooding was generated in Ireland, this is nearly an average of one per year. It is interesting to note that all but 2 of the periods occurred between late autumn (October) and early spring (March). The inundation levels associated with the storm systems of Table 3 vary from the threshold level of 0.5 m to a couple of metres. Sources were scored according to their reliability, from 'A' to 'C'. A level 'A' source is an official technical report, level 'B' indicates a report in a newspaper article, while a level 'C' source is an eyewitness account in a newspaper article.

### 4.3.2 1942 Co. Kerry

A severe storm swept over Co. Kerry during the week of the 12 December 1942 with the "Highest Seas since 1928". Many boats were destroyed and the storm was described as what may be "the worst ever experienced", with flooding and the seawalls were "completely demolished", "badly breached" or "washed way". One house was flooded with sea water 0.6 m (2 ft) deep. A well known landmark, a large rock, disappeared. The lighthouse lamp was put out by huge seas and high winds. Large quantities of seaweed were strewn across the streets and the beaches were covered with fish. Damage to fishing boats and nets were caused by "mountainous seas and very high tides".

It was believed that sheep were blown off the cliffs in one area, while it was reported that a "tidal wave" washed seven cattle and sixty sheep off an island near the Derrynane coast. There was a narrow escape by a 73 year old man who swam to safety in Kenmare Bay when his boat was swept from her mooring and smashed into pieces. Unfortunately his little dog was found with the wreckage the following day. (*The Kerryman* 19 December 1942)

The description of high tides and tidal wave suggests that this event was a storm surge.

### 4.3.3 1961 Hurricane Debbie, West of Ireland

The centre of Hurricane Debbie passed close to the west coast of Ireland on 16 September 1961, bringing winds gusting over 171 km/hr (110 mph). There was extensive damage to property, with 11 deaths attributed to the storm (Met Éireann, 2016a), while newspapers reported 15 deaths (*Irish Independent*, *The Irish Times* 18 September 1961).

A French weather ship reported wind of 114 mph [183 km/hr] and 45 ft [13.7 m] waves in the Aran Island Region. 7 small craft were sunk at their moorings at the Galway docks and one swept away. (*Connacht Tribune* 23 September 1961). At Lough Sheelin, Kilnaleck, boats were carried inshore, some as far as 50 yards [45.7 m]. (*The Anglo-Celt* 23 September 1961)





**Table 3.** Coastal flooding in excess of 0.5 m 1961 - 2005 in the North East (NE), South East (SE), South (S), South West (SW), West (W), North West (NW). The reliability of sources are labelled from 'A' to 'C': where 'A' indicates an official technical report, 'B' indicates a report in a newspaper article, and 'C' is an eyewitness account in a newspaper article.

| Flood Date | Region | Location | Inundation | Source | Reliability |
|---|---|---|---|---|---|
| 22/10/1961 | NE | Glin, Limerick | 1.2m | The Kerryman | B |
| 24/10/1961 | W | Kinvarra, Galway | 1.8m | Connacht Tribune | B |
| | W | Galway city and Salthill | 1.2m | Connacht Tribune | B |
| 07/03/1962 | NE | Quay Street Dundalk, Louth | 0.6m | The Irish Press | B |
| | NE | Boyne Road Drogheda, Louth | 0.6m | The Irish Times | B |
| | SE | Main St, Wexford | 0.6m | The Irish Press | B |
| 17/11/1963 | NE | Clontarf, Dublin | 0.6m | The Irish Press | B |
| 22/12/1968 | SW | Ballynacally - Ennis Road | 1.2m | Irish Independent | B |
| 02/02/1972 | S | The waterfront and railway line, Wexford | 0.6 − 1.2m | Irish Independent & The Irish Press | B |
| | S | The quayside, Waterford | 0.6m | Irish Independent | B |
| | SE | Strand Road, Bray, Wicklow | 0.9m | The Irish Press | B |
| 08/01/1974 | S | Bridgetown, Wexford | 0.6m | Cork Examiner | B |
| 10-11/01/1974 | W | Road at Clogher, near Belmullet | 1.2m | Connaught Telegraph | B |
| 10-11/01/1974 | NW | Caltragh, Sligo | 1.5m | Sligo Champion | C |
| 10/01/1974 | S | Cork city | 0.9m | The Irish Press | B |
| 10/01/1974 | S | Cobh, Cork | 0.6m | The Irish Press | B |
| 10/01/1974 | S | Waterford city | 0.6m | The Irish Press | B |
| 11/01/1974 | S | Ballinacurra, Cork | 1.2m | Cork Examiner | C |
| 12-13/01/1974 | W | Frenchville, Galway | ≥ 0.6m | Connacht Sentinal | B |
| 11/11/1977 | W | Galway city | 0.6m | The Irish Press | B |
| 12-13/12/1981 | S | Timoleague - Courtmacsherry Road | 0.9m | The Southern Star | B |
| 13/12/1981 | NE | Clontarf road, Dublin | 1.2m | The Irish Press | B |
| 13/12/1981 | SE | Bray, Wicklow | 1.2m | The Irish Press | B |
| 14/12/1981 | SW | Limerick city | 0.9m | Cork Examiner | B |
| 18/12/1983 | S | Clonakilty, Cork | 0.6m | Cork Examiner | B |
| 07/04/1985 | S | Cork city | 0.9m | Irish Independent | B |
| 04/12/1986 | NW | Rossbeg, Donegal | ≥ 0.6m | Mayo News | B |
| 21/10/1988 | S | Cork-Killarney road | 0.6m | Cork Examiner | B |
| 01/03/1989 | SW | Spanish Point, Clare and | 1.2m | Irish Independent | B |
| 08-09/03/1989 | W | Spanish Parade, Galway | 0.6m | Galway City Tribune | B |
| 09/03/1989 | SW | 5 Mile Bridge and Castlecountess - Ballymullen Road, Kerry | 0.6m | Cork Examiner | B |
| 14/12/1989 | NE | Blackrock and Salthill train stations, Dublin | 0.6m | Irish Independent | B |
| 25/08/1992 | NW | Derrybeg, Donegal | 1.5m | Donegal News | B |
| 25/08/1992 | NW | Annagry, Donegal | 0.9m | Donegal News | C |
| 17/01/1995 | SW | Limerick city | 0.6m | Limerick Leader | B |
| | W | Claddagh, Galway | 0.6 − 0.9m | Connacht Tribune | B |
| | W | Grattan Park, Galway | 0.9m | Galway City Tribune | B |
| 10/01/1997 | W | Galway city | ≥ 0.6m | Connacht Sentinal | B |
| 09/02/1997 | SW | Corbally, Limerick | 0.9m | Limerick Leader | B |
| 10/02/1997 | W | Galway city | ≥ 0.6m | Connacht Sentinal | B |
| 04-08/01/1999 | W | Achill Island, Mayo | 0.5m | Irish Examiner | B |





| 01/02/2002 | SE | New Ross, Wexford | ≥ 0.6m | Irish Independent | B |
| | SE | Bray, Wicklow | 1m | OPW | A |
| | NE | Stella Gardens, Dublin | 2.1m | Dublin City Council | A |
| | SW | Quilty, Clare | 2.1m | Office of Public Works (eyewitness account) | C |
| | SW | Ballylongford, Kerry | 0.9m | Office of Public Works | A |
| | NE | Mornington District, Drogheda | 0.6m | Office of Public Works | A |
| | NE | Marsh South, Dundalk | 0.5m | Office of Public Works | A |
| 21/11/2002 | S | Blackpool and Togher, Cork | 0.6 − 1.2m | Irish Examiner | B |
| 21/11/2002 | S | Cork city | ≥ 0.6m | Irish Independent | B |
| 27/10/2004 | SE | Wexford Quays | 1.2m | Wexford County Council and The Irish Times | A |
| | S | Crosshaven road, Cork | 1.6m | Cork City Council | A |
| | S | Dungarvan, Waterford | 1.5m | The Irish Times and Irish Independent | C |
| | S | Dunmore East, Waterford | 1.5m | The Munster Express | B |
| | S | Cork city and Carrigaline | 1.2m | Irish Examiner | B |
| | S | Waterford city | 0.9m | Irish Examiner | B |
| | S | Dungarvan, Waterford | ≥ 0.6m | Irish Examiner | B |
| 11/01/2005 | SE | Tralee, Kerry | ≥ 0.6m | The Kerryman North Edition | B |
| 03/12/2006 | W | Crossmolina, Mayo | 0.9m | Irish Independent | B |

### 4.3.4 1962 Youghal, Co. Cork

A major storm caused severe high tides and flooding in Youghal, Co Cork on the 7 March 1962 and surrounding days. The town was severely damaged and the sea wall collapsed. The waves on the beach, "revealed rarely seen peat masses... evidence of our past connection with Yew Trees".

5     There was also reports of flooding and heavy seas causing damage at Rosslare, Co. Wexford, on the train line at Wexford town and between Greystones and Kilcoole, Co. Wicklow. Flooding and huge waves also hit Bray, Co. Wicklow and Clontarf, Co. Dublin, and a small landslide at Bray Head. Heavy seas at Kilmore Quay, Co. Wexford tore a trawler from its moorings. At Skelligs Rock lighthouse off Co. Kerry, a wall, railings and roadway were washed away by "mountainous seas". In addition, a coaster was driven ashore by strong winds and huge waves at Ards peninsula Co. Down. (*The Irish Times* 9 March 1962, RTÉ 10   (2016c): *Aftermath of Storm Youghal 1962*).

    The ICPSS (The OPW and RPS, 2013) don't include this event in their reports, however there was clearly significant coastal flooding associated with it. Therefore, flooding in excess of 0.5 m is shown in Table 4 in the same style to Table 3.

### 4.3.5 1974 Storms

The weather in January 1974 was mild, wet and stormy as the North Atlantic and a large section of Northwest Europe was under 15   a "sway of complex low pressure area". An extreme storm during the 11-12 January caused extensive damage. In particular full moon spring tides combined with wind and low pressure caused damage to low lying coastal farms and houses. "Roads were blocked by seaweed, rocks or other debris and in some sections destroyed". Many small boats sank and larger boats were torn from their moorings. At Inisboffin, off Donegal, waves were sweeping through the centre cutting it in two temporarily. Also,





**Table 4.** Coastal flooding in excess of 0.5 m on 7th March 1962. The reliability of sources are labelled from 'A' to 'C': where 'A' indicates an official technical report, 'B' indicates a report in a newspaper article, and 'C' is an eyewitness account in a newspaper article.

| Location | Inundation | Source | Reliability |
|---|---|---|---|
| South Terrace, Cork | 2m | Irish Independent | B |
| Tragumna, Cork | 2.1m | The Southern Star | C |
| Crosshaven road, Cork | 1.8m | The Southern Star | B |
| Middleton, Cork | 1.5m | The Southern Star | B |
| Carrigaline, Cork | 0.9m | Irish Independent and The Southern Star | A |
| Kinsale, Cork | 0.9m | The Southern Star | B |
| Bantry and Union Hall, Cork | 0.6m | The Southern Star | A |
| Waterford city | 0.9m | Munster Express | B |
| Kilshannig, Kerry | 1.5m | The Kerryman | C |

a "freak event" at Crushoa, in south Galway occurred on the evening of the 11th. Seaweed farmers noticed the tide "reversed for about an hour and then returned with full force and vigour not discernable in its earlier movement". (Met Éireann (2016a), *Connaght Tribune* 18 January 1974).

The descriptions given here imply that this event was a storm surge.

### 4.3.6 1976 Storm

A storm on 2nd January 1976 caused widespread damage in Ireland. A depression intensified considerably off the north coast of Ireland taking people by surprise and generating storm force winds in many areas. At Limerick, the tide was just 7.6 cm short of its record level (68.6 cm). The coastal town of Ballina, Co. Mayo reported flooding, though it is not clear if this was coastal flooding or due to the river bursting its banks. The sea at Kinvara, Co. Galway "cascaded across the quays" and the tides "swept over the road in two places" nearly cutting off the town. (Met Éireann, 2016a).

The high tides and coastal flooding clearly suggest this was a storm surge event.

### 4.3.7 1986 Storms and Hurricane Charley

August 1986 brought a succession of storms and flooding to Ireland. On the 5th a vigorous depression approached from the southwest bringing record rainfall to Valentia, Co. Kerry and west Cork. In Tralee, Co. Kerry the rain combined with a high tide caused a subterranean river under the town to flood. At Bantry, Co. Cork shops were in 3 feet of water at high tide.

Just 10 days later, tropical storm Hurricane Charley appeared off South Carolina and intensified to a hurricane as it tracked north-northeast. On the 18th Charley began to decline as it headed out into the Atlantic, and by the 22nd it was no more than an ordinary depression. However, the following day it began to deepen rapidly and by the 24th it had clearly developed into a



separate depression. It tracked about 500 kilometres southwest of Kerry and continued south of Ireland over the coming days causing major flooding and damage. In fact, it was the worst flooding in Dublin for 100 years and described at the time as one of the worst storms in living memory. It seems that the flooding associated with Hurricane Charley was associated with heavy rain rather than a storm surge. (Met Éireann (2016a), *The Irish Times* 6 August 1986)

### 4.3.8   2002 Co. Dublin

Atlantic depressions brought frequent rain and gales across the country in February 2002, causing rainfall levels to be twice the monthly normal in many parts of the country. This led to flooding in many areas. However, a deep depression that passed northeast of the country on the 1st March brought exceptionally high tides, especially in the Irish sea. Dublin was worst affected with its highest tide measured for over 80 years. Sea defences failed and rivers and canals burst their banks causing major flooding in parts of the city. (Met Éireann, 2016a)

## 4.4   Tsunamis

### 4.4.1   1942 Ventry habour, Kerry

A "tsunami" was observed at Ventry harbour on a calm, sunny day in September 1942. During a "series of huge waves", "you could see the bottom of the ocean floor". It is said that sheep moved to higher ground before the arrival of the wave and locals were picking up fish from the fields after the event. The event seemed to only affect Ventry bay region and not the surrounding harbours on the other side of the peninsula. This might indicate that resonance occurred in the bay due to its shape. (*Eyewitness account provided by Dunbeg Fort and Visitor Centre*)

### 4.4.2   2011 South Coast

The National Oceanic and Atmospheric Administration (2016) Tsunami database records a questionable tsunami on 27 June 2011 in Plymouth, England with run up values recorded at Newlyn, Plymouth, Portsmouth and St Michael's Mount. The tide gauge record at Castletownbere, Wexford and Ballycotton shows evidence of this wave in Figure 7. The tsunami cause is given as meteorological on the NOAA database, however, a BBC News report on 29/06/2011 suggests it was due to a sub-marine landslide.

## 5   New Events 2012 - 2016

## 5.1   Storm Waves

### 5.1.1   2014 North Co. Wexford

A storm on the 13 January 2014 damaged a stretch of the North Wexford coastline which was described as "some of its worst damage in decades". In particular, a section of a house in Ardamine was "left hanging in the air, when the cliff face below




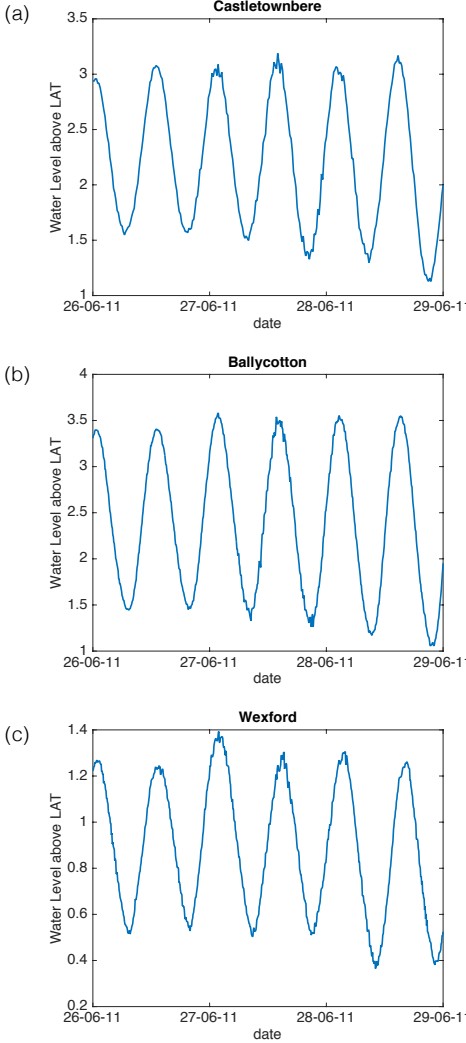

**Figure 7.** Tide gauge data from 26th-29th June 2011 at (a) Castletownbere (SW coast), (b) Ballycotton (S coast) and (c) Wexford (SE coast), small oscillations are clearly visible.



was washed away". In addition, part of a car park in Cahore, a beach access road at Ardamine and a walking trail in Courtown Woods were washed away. One local described "a high tide, and the wind in the right direction with a big swell", creating the very damaging conditions. (*The Wexford People* 14 January 2014)

### 5.1.2   2014 West Co. Cork

Two men died while out walking along the coastline in west Cork. Apparently the two men "had hoped to take photos of stormy seas striking rocks" by a local lighthouse. The route known locally as 'Poet's Path' is very exposed in parts, to the wind and sea. Gardaí (Irish police) assumed that freak waves and gusts of up to 130km/hr caused them to get into difficulty. (*Irish Independent*, 11 February 2014).

### 5.1.3   2014 Belderrig, Co. Mayo

The front of a stone built boathouse was destroyed at Belderrig near Belmullet, Co. Mayo, after a large wave hit it on the 10 December 2014. The boathouse sits approximately 10 m above sea level. Nearby, at about 15 m above sea level, a sizeable boulder was found overturned, and about 1 km from the boathouse, at approximately 30 m above sea level, large rocks ($\sim$ $40-50$ kg) were also found strewn about. One of the locals noticed that the sea withdrew dramatically after one of the large waves hit the coast.

Nearby at Portacloy, the sea state caused a World War 2 look out post to collapse on the same day. A photo of the sea condition in the area on this day are shown in Figure 8.

The M4 buoy located off the coast, Northeast of Belderrig, recorded a maximum wave height of 21.5 m on the same day, with a corresponding significant wave height of 14.5 m (see Figure 9). This is not classified as a rogue wave, however the waves hitting the coast of Co. Mayo on this day did significant damage. A local man pointed out that if the large waves had hit the coast at high tide, the damage could have been far worse. (*Personal contact with Seamus Caulfield and Gretta Byrne*)

### 5.1.4   2015 Hook Head, Co.Wexford

A 14 year old girl died after she and three other teenagers were swept into the sea off Hook Head, Co. Wexford during a heavy swell in the aftermath of Storm Desmond on the 6th December 2015. The four were part of a scouts group expedition. It was reported that they were walking on the rocks along the foreshore when a "rogue wave" dragged them out to sea. (*The Irish Times*, 7, 10 December, 2015).

### 5.1.5   2016 Culleton's Gap, Co. Wexford

The sand dunes at Culleton's Gap (The Raven), near Curracloe, Co. Wexford were altered after gales on the 20th August 2016 [1]. Trees that line the beach were damaged, sand was shifted and tree roots were exposed (see Figure 10). Locals say that this

---

[1]*Personal communication with James Herterich*





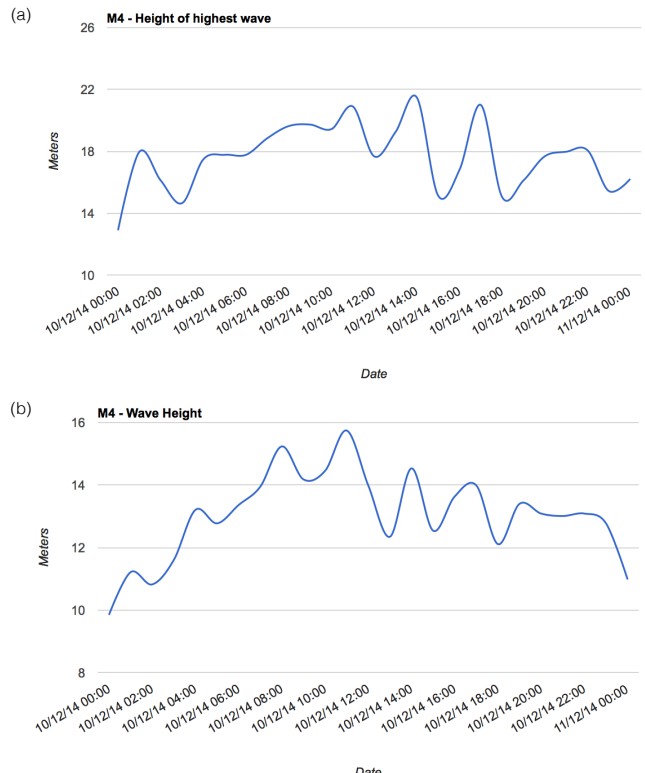

**Figure 8.** The sea condition as viewed from the platform at the Céide fields, Co. Mayo on the December 10th 2014.

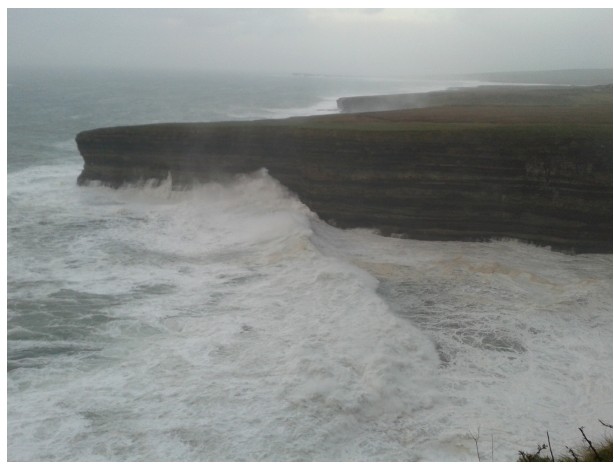

**Figure 9.** M4 records of (a) maximum wave height (m) and (b) significant wave height (m) on the 10th December 2014.




is a common phenomenon in the region, particularly with strong easterly winds. Presumably it is a combination of wind and waves that alters the dunes here.

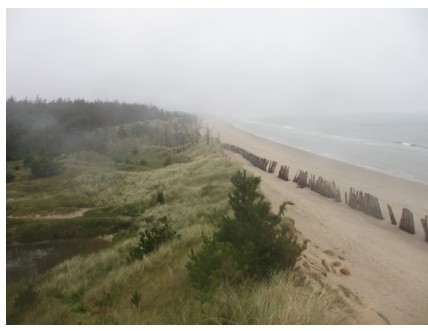 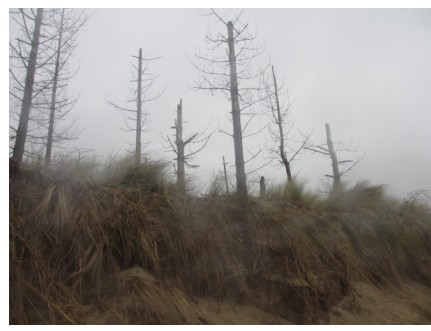 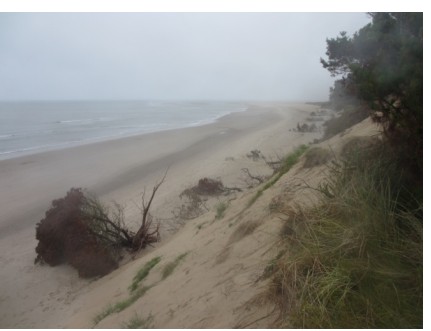

**Figure 10.** Pictures of the sand dunes at Culleton's Gap, Co. Wexford taken on 18 September 2016 approximately one month after the gales. (Source: *James Herterich*)

### 5.1.6   2016 Irish Sea

A passenger ferry was forced to spend the night in the Irish sea during a storm on the 21st - 22nd November 2016. The ferry was bound for Fishguard, Wales travelling from Rosslare, Co. Wexford and had two failed attempts of docking before deciding to remain off the Welsh coast overnight. The ferry finally docked at 11 am on the 22nd. (The Irish Times 22 November 2016)

## 5.2   Rogue Waves

### 5.2.1   2013 Ballyrean, Co. Clare

On the 6th October 2013 a Latvian national was washed off the rocks by a "strong wave as he attempted to fill a bucket with water" at an area known locally as the Fisherman's Climb, near Ballyrean, Co. Clare. (*Irish Independent*, 6 October 2013, *Irish Times*, 8 October 2013).

### 5.2.2   2013-2014, Waverider Buoy, Killard, Co. Clare

A rogue wave was observed by a Waverider buoy near Killard deployed by the ESB, on the 28th January 2014 with a maximum trough-to-crest (upcrossing) value of 26.45m (Fedele et al., 2016). Analysis of the raw data shows that a rogue wave with a crest-to-trough/downcrossing value of 33.96 m was possibly recorded on 26th January 2014 at 22:30. See Figure 11. Note that there were 7 rogue waves greater than 20 m identified on this day, using both the upcrossing and downcrossing method, though
2 were part of the same wave packet. In addition there were 3 rogue waves greater than 20 m identified the following day, 27th January 2014 but none on the 28th.



Furthermore, multiple rogue waves greater than 20 m have been recorded by this buoy between November 2011 and January 2015. It should be noted that the buoy was out of action Feb - Jul 2012 and Jan - Sep 2013 for various reasons. A total of 20 waves (on 7 days) were identified using upcrossing and downcrossing methods. For each of these days, the rogue wave with maximum wave height is plotted in Figure 12. Note that some values are missing due to the devices recording being irreparable. (The sensors on the Waverider buoy can only record a displacement of $\pm 20.48$ m. Waves may actually have been higher, just not recordable.)

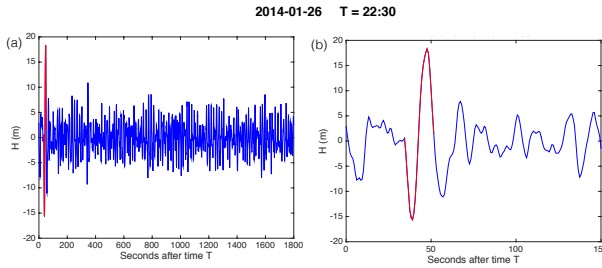

**Figure 11.** 33.96 m wave recorded at Killard ESB Platform on 26th January 2014 (a) between 22:30 and 23:00 and (b) zoomed in. The wave as identified by its crest-to-trough is highlighted in red.

### 5.2.3 2015 ADCP Measurements

Flanagan et al. (2016) deployed a state-of-the-art Sentinel V Acoustic Doppler Current Profiler (ADCP) off the west coast of Ireland near Killard, Co. Clare from the 9th February to 1st May 2015. Their aim was to gather accurate wave measurements in extreme conditions. Of the 750,000 waves recorded and analysed, 13 were rogue waves (approximately 1 in every 60,000) and only 3 of these waves were greater than 4m and none exceeded 10m. A similar study of field measurements (Christou and Ewans, 2014) measured 3649 rogue waves in 122 million individual waves (approximately 1 in every 30,000).

A number of large waves with amplitudes close to 20m were recorded on the 22nd February, see Figure 13. This shows that large waves of interest are not always classified as rogue waves.

Note that the ADCP is deployed in an area with water depth $h = 36.5$ m, therefore we would expect that waves here would not exceed heights of $H = 0.55h = 20$ m.

### 5.2.4 2015 The Wormhole, south of Dún Aonghasa, Inis Mór, Aran Islands

On the 8th of April 2015 a student was swept off a cliff face and into the sea by a giant wave at the renowned Wormhole (a rectangular shaped pool at the bottom of cliffs) on Inis Mór. Luckily people in the area were able to go to her aid and saved her using a make shift rope. Video footage can be seen of the occurrence on youtube (Brian Smith Music, 2015). (*The Irish Times* 15 April 2015).



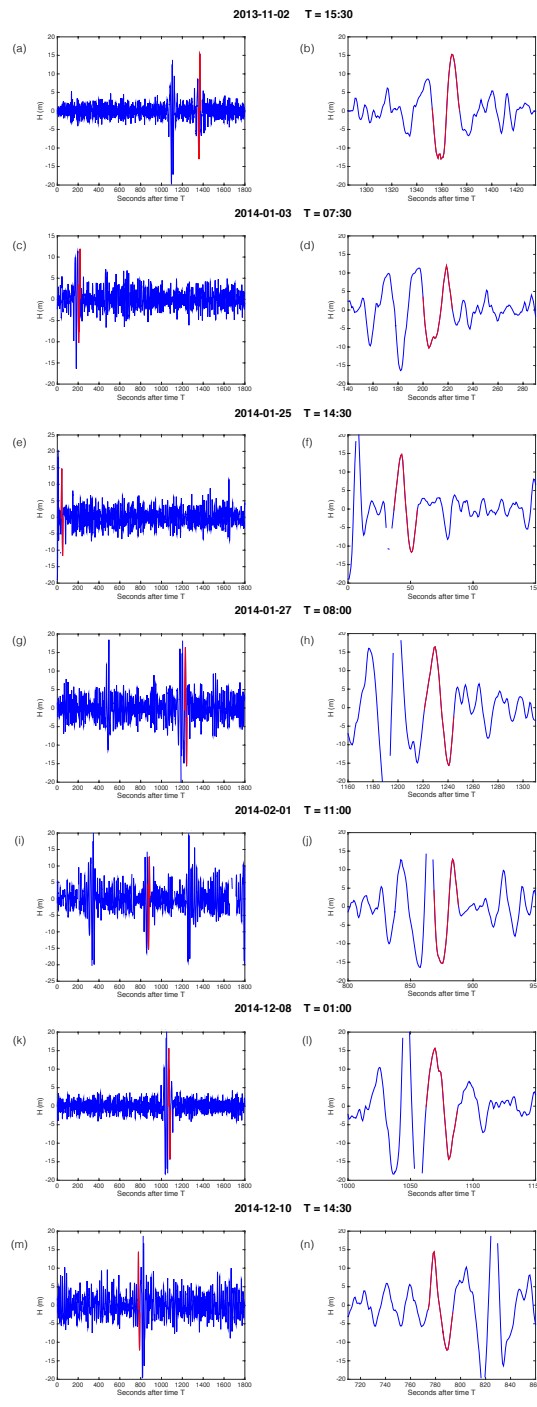

**Figure 12.** Large rogue waves recorded at Killard ESB Platform. Wave height recordings for the 30 minute duration when wave occurred (left column) and zoomed around the time of event (right column). The waves are highlighted in red. (a), (b) 02/11/2013, 15:30. (c), (d) 03/01/2014, 07:30, (e), (f) 25/01/2014, 14:30. (g), (h) 27/01/2014, 08:00. (i), (j) 01/02/2014, 11:00. (k), (l) 08/12/2014, 01:00. (m), (n) 10/12/2014, 14:30.





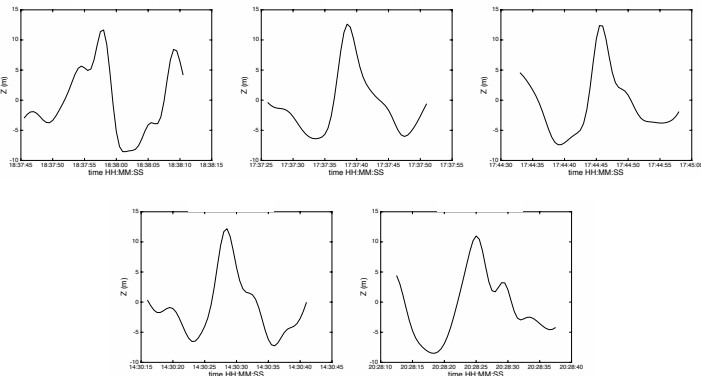

**Figure 13.** Large waves recorded by an ADCP near Killard, Co. Clare on the 22nd February 2015.

### 5.2.5  2015 Baltimore, Co. Cork

On the 30th June 2015 four people were hit by a rogue wave while fishing near the Beacon in Baltimore, only one survived. They were initially hit by a couple of waves and then a "big wave" dragged two of the group out to sea. The other two were pushed up against the rocks, one then jumped into the sea in an attempt to rescue the others but sadly all three drowned. (*The*

*Irish Times* 26 April 2016).

### 5.2.6  2016 Ballyrean, Co. Clare

A woman drowned after being swept out to sea by a "rogue wave" while fishing with friends in Ballyrean, south of Fanore, Co. Clare on 10th July 2016. Conditions at the time were harsh with winds reaching gale force 6-7. (*Independent.ie* 10 July 2016, *The Irish Times* 10 July 2016).

### 5.2.7  2016 Storm Jake

Storm Jake in late February/early March 2016 brought severe weather to Ireland. On the 2nd March at the Belmullet Berth B wave buoy, a 30.96m wave was recorded with a significant wave height of 8.8 m, classifying it as a rogue wave. See Figure 14. However, the buoys can only be considered reliable within a certain range of displacement from mean sea level, so very large wave recordings like this should be treated with caution. Although, the nearby Belmullet Berth A wave buoy also recorded a

very large wave at around the same time (Figure 14) giving more confidence in the recording.

### 5.2.8  2016 Belmullet Berth A

On the 7th August 2016, the Belmullet Berth A wave buoy recorded a rogue wave, in excess of 13 m (significant wave height of approximately 6 m). This was during a period of unusually windy conditions for the summer season.



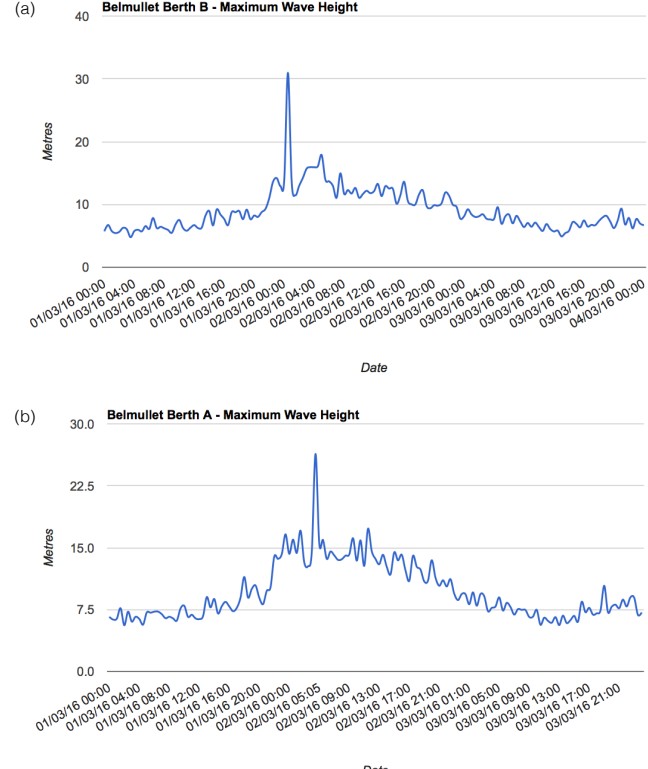

**Figure 14.** Maximum wave height at (a) Belmullet Berth B and (b) Belmullet Berth A wave buoys on 1st - 3rd March 2016.

## 5.3 Storm Surges

The winter of 2013-2014 was severely affected by a large number of storms due to the atmospheric jet stream extending right over Ireland carrying successive storms. This exceptional weather combined with high tides resulted in serious coastal damage and widespread flooding (Met Éireann, 2016a).

5     In particular, high spring tides on the 3rd January and 1st February coincided with storms on the 2nd - 3rd January, 5th- 6th January and 1st February causing extensive storm surges and damage around the country. These events are described in more detail below.

### 5.3.1 2014, January Storms

Storms at the end of December caused some coastal flooding in Cork, Wicklow and Dublin (*Irish Independent* 2 January 2014)
10  but the storms on the 2nd - 3rd and 5th - 6th January 2014 had far greater consequences. Tide gauges at Rossaveal, Co. Galway and Clarecastle, Co. Clare on the 3rd and 6th recorded large surges combined with a high spring tide causing extensive damage and flooding along the western coastline (Moroney, 25 September 2015).



On the 2nd and 3rd there was extensive flooding around Galway city and Salthill, Co. Galway, while six cars were swept off the pier at Cleggan. In Lahinch, Co. Clare, the promenade was destroyed by the storm surge, while many other coastal regions in Co. Clare coastline were flooded and damaged. In Westport and Ballina, Co. Mayo there was severe coastal flooding and damage. (*Sunday Independent* 5 January 2014)

Over 0.6 m of flooding was reported in Foynes, Co. Limerick and severe damage along the Kerry coastline, particularly at Rossbeigh Beach, Ballybunion, Dingle and Ballinskelligs. (*The Kerryman* 8 January 2014)

Storm surges also effected other parts of the country. In Cork city, flood levels reached 0.6 m on the 2nd and coastal flooding in Northern Ireland around Belfast, Newry and Coleraine. Some coastal flooding also occurred in Dublin, Waterford and Kerry. (*Irish Independent* 3 and 5 January 2014, *Belfast Newsletter* 4 January 2014)

The storm on the 5th - 6th January exasperated damage, particularly along the Galway coast with "chunks of coastline" ripped off. Many coastal graveyards were affected, infrastructure on the Aran Islands and Inisboffin were badly damaged and parts of road were washed away along the Connemara coast. There was also severe tidal flooding on Achill Island. (*Galway City Tribune* 10 January 2014, *Connaught Telegraph* 7 January 2014)

At Blacksod lighthouse, Co. Mayo, the helipad had to be closed after it was submerged in 0.6 m water and there was damage
to paving stones and surrounding wall (Sweeney, 2014).

### 5.3.2   2014, February Storm

Another storm on the 1st February 2014 caused more coastal flooding and damage across the country. Coastal flooding was worst in Co. Cork, particularly in Cork city, Clonakilty and Kinsale. In Co. Kerry there was over 0.75 m flooding in homes around Ballybunion. Floods also hit the Maharees area causing damage to Scraggane Pier road, Castlegregory beach and the
sand dunes at Stradbally beach. Areas of Limerick, Waterford and Galway (see Figure 15) were also effected by flooding. (*Irish Independent* 5 February 2014, *The Kerryman* 5 February 2014, *The Southern Star* 8 February 2014).

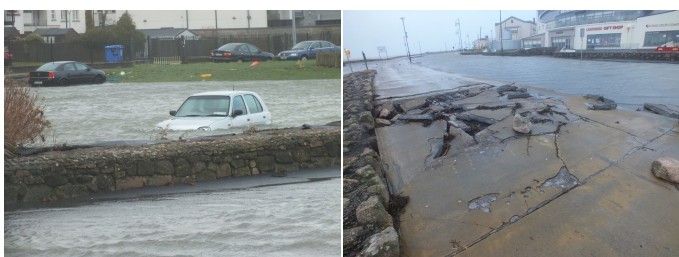

**Figure 15.** Photos of Galway on 1st February 2014 (credit: Emiliano Renzi)

### 5.4   Tsunamis

No tsunamis were recorded for Ireland between 2012 and 2016.




### 5.5  New Surf Waves

#### 5.5.1  Potential big wave surf spot, The Atlantic Ocean.

Big wave surfer Andrew Cotton believes that a reef off the west coast of Ireland has the potential to produce some of the biggest surf waves in Ireland. Andrew first noticed that the region had potential in 2010, and in 2013 he saw approximately 15 m (50
ft) waves there, and he believes it could easily get to 18 m (60 ft). Since 2015 he has been watching the charts and conditions waiting for the perfect weather conditions, tide and swell (Red Bull TV, 2016).

#### 5.5.2  Riley's wave, Co. Clare

The discovery of Riley's wave in the late 2000s is attributed to Mickey Smith. The location of big surf waves along the Irish coast can be difficult to locate and generally require local knowledge and precise directions. See a picture of surfer climbing up
from Aileen's wave in Figure 16. In the case of Riley's wave you "trek across a few fields that turn boggy in winter and walk through a herd of cattle... jump a few watery ditches and climb a gate and then begin to walk across the long rocky ledge which is lethally slippery all year round but farcical in winter, when the surface freezes over... Rock falls, from pebbles to boulders, happen regularly; the platform is strewn with smashed-up stone. About halfway across lie the dried-out hides of a horse and foal which fell from the headland a couple of years ago. The rock shelf has collapsed in the middle so you have to skip across
a narrow ledge to get across to where the wave breaks." (*The Irish Times* 9 October 2012).

#### 5.5.3  Bumbaloids, Co. Clare

Originally a bodyboarding location the wave was first surfed by Feargal Smith and Mickey Smith in April 2007 (Smith, 2012). yoSurfer (2016) pinpoints Bumbaloids between Doolin and Dunagore, and rates the size of the waves between $0.9 - 3.7$ m $(3 - 12$ ft$)$. See a picture of sunset at Bumbaloids in Figure 17.

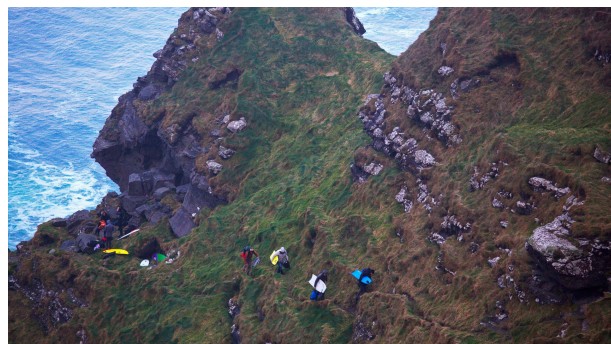

**Figure 16.** Surfers hike back up a cliff after surfing Aileens wave, Co. Clare (Craig, a).





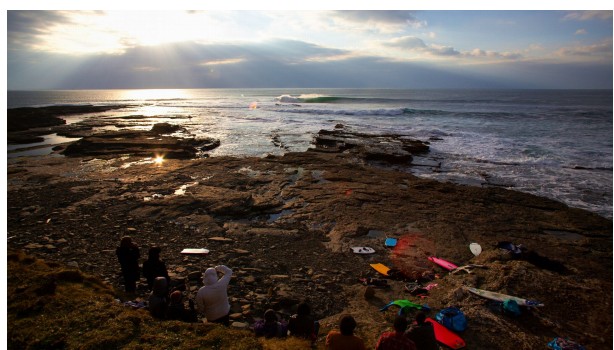

**Figure 17.** Sunset at Bumbaloids, Co. Clare (Craig, b).

## 5.6 Buoy Network

The Irish Marine Institute manage a group of offshore buoys deployed around the Irish coast that monitor weather and oceanographic conditions in real-time. Figure 18 shows the distribution of rogue wave and large waves (> 10 m) recorded for buoys M2 - M5 during the period for which data is available for each buoy between March 2011 and June 2016. Notice that waves
5  classified as rogue waves are not often very large. Also, there is a much greater number of large (non-rogue) waves than rogue waves, indicating that more often than not a very large wave is not classified as a rogue wave.

Table 5 shows the number of rogue waves and large waves for buoys M2 - M5 (during the period for which data is available for each buoy between March 2011 and June 2016). Again this highlights the difference between numbers of rogue wave and large (non-rogue) waves, particularly for M3 and M4 which lie off the West coast of Ireland. Notice that M2 and M5
10  buoys don't record many of either wave type, these buoys are located off the East coast and Southeast coast of Ireland in more sheltered waters.

**Table 5.** The number of rogue waves and large waves recorded for buoys M2 - M5 between March 2011 and June 2016 (M2: 31/03/11 - 01/07/16, M3: 01/06/12 - 01/07/16, M4: 07/06/11 - 01/02/16, M5: 23/01/12 - 01/07/16).

| Buoy | Rogue Waves > 10 m | Non Rogue Waves > 10 m | Non Rogue Waves > 20 m |
|------|-----|-----|-----|
| M2 | 0 | 0 | 0 |
| M3 | 4 | 697 | 2 |
| M4 | 5 | 1054 | 7 |
| M5 | 2 | 7 | 0 |





**Figure 18.** Number of rogue waves (left column) and number of large waves greater than 10 m (right column) recorded for each buoy (a), (b) M2; (c), (d) M3; (e), (f) M4; (g), (h) M5; for data available between March 2011 and June 2016 (M2: 31/03/11 - 01/07/16, M3: 01/06/12 - 01/07/16, M4: 07/06/11 - 01/02/16, M5: 23/01/12 - 01/07/16).




## 6 Boulder Deposits

Evidence of boulder deposits due to waves can be found in numerous areas around Ireland (Hansom and Hall, 2009; Williams, 2010; Cox et al., 2010). A review of possible mechanisms for this type of boulder movements is given in O'Brien et al. (2013), though there is no consensus in determining the size of waves at the coast and the masses they can carry. Boulder deposits can be used as indicators of the impact and sheer force of the ocean. However, there is a lack of data quantifying the effects. In 2014 Dr. Rónadh Cox and her group from Williams College, MA, USA, collected an important dataset on the Aran Islands, off the West coast of Ireland documenting the movement of $> 1000$ boulders.

Below we list some more recent reports of boulder deposits due to extreme waves in Ireland.

### 6.1 1991 Clifden, Co. Galway

Renvyle House hotel in Clifden suffered severe damage from a storm in early January 1991. It was reported that wave and boulders lashed the house. The owner described how a "tidal wave had hurled rocks and stones". (*The Irish Independent 7th January 1991*)

### 6.2 2014 Lahinch, Co. Clare

A storm on the 2nd-3rd January 2014 generated a severe storm surge that caused "Huge boulders" to be thrown across the car park at Lahinch promenade, Co. Clare. (*Sunday Independent* 5 January 2014). Similar damage occurred at Rossbeigh Beach, with "tarmac rocks and boulders" distributed around the area after the storm. (*The Kerryman* 8 January 2014)

### 6.3 2014 Brandon, Co. Kerry

A storm surge on the 1st February 2014 caused damage to the pier at Brandon, Co. Kerry as the sea "threw up massive boulders and debris". (*The Kerryman* 5 February 2014)

### 6.4 Horse Island, Co. Mayo

A local man [2] has noticed that there is dramatic movement of large rocks on Horse Island, Co. Mayo. It is estimated to have occurred at approximately 30 m above sea level.

## 7 Climate Change

Global warming will lead to a rise in mean surface temperatures of between $0.18°$ C and $4.8°$ C by the late 21st century (IPCC, 2013). This will result in changes to global circulations, particularly the atmospheric jet stream, which is a major driver of mid-latitude weather.

---

[2]Personal contact with Seamus Caulfield





Although climate models mostly agree on a poleward shift of the jet stream in response to anthropogenic forcing, there is still considerable spread between different model projections (Woollings and Blackburn, 2012) and uncertainty in how this will affect weather. In particular, there is low confidence in projecting changes in the Northern Hemisphere winter storm tracks, especially in the North Atlantic Basin (IPCC, 2013; Gallagher et al., 2016).

Ireland is strongly influenced by the position of the jet stream as it generally coincides with the path that storms will take as they pass over or near Ireland. Various studies have downscaled climate projections over the 21st century to determine how Ireland's climate is expected to change, the results of these studies are combined in Gleeson et al. (2013).

Some of the expected changes that could influence extreme wave events in Ireland are outlined below:

– Increase of cold continental air outbreaks during winter.

– Winters to become wetter (up to $14\%$).

– Increase in frequency of heavy rain events in winter (up to $20\%$).

– Increase in energy content of the wind (up to $8\%$).

– A small decrease in mean wave heights, while in winter and spring, storm wave heights are likely to increase in the north and northwest.

These results indicate that, in general, the severity of winter weather is expected to increase indicating that extreme wave events may increase.

However, actual wave climate projections for Ireland have been carried out by Gallagher et al. (2016). This is the first study of its kind providing the highest resolution wave projection data set available for Ireland. They found that 10 m wind speeds over the North Atlantic Ocean ($5 - 75°$N, $0 - 80°$W) are expected to decrease by the end of the century (in means up to 3%

and extremes up to 14%). They also predict an overall decrease in mean and extreme (up to 15%) annual, winter and summer significant wave heights around Ireland. These results indicate that extreme wave events are expected to decrease in the future. This seems at odds with the prediction that the winter weather is expected to be more severe. However, it should be noted that the results from Gleeson et al. (2013) are based on mid-century outcomes, while the Gallagher et al. (2016) study uses a different forcing that represents the end of the century.

Further work is required increase confidence in these results since uncertainty remains in the future position of the jet stream, in particular in the North Atlantic (IPCC, 2013; Gallagher et al., 2016), and subsequently the projected changes in wave heights. Additionally, recent research suggests that the effect of the North Atlantic Oscillation (NAO) should be taken into account when considering future extremes, as a positive phase may in fact enhance extremes of significant wave height (Gleeson et al., 2017).

**7.1 Coastal Erosion**

Ireland has approximately 6000 km of coastline. The Winter storms of 2013-2014 caused huge damage and erosion along this coastline. It is still unclear whether these storms can be attributed to climate change, but Murphy (2014) found that wave



conditions that are only expected to occur once a year were exceeded 7 times in just one month during this time. Murphy (2014) warns that if severe storms like these become more frequent, Ireland will need to come up with a strategy for coastal erosion. Further, a predicted sea level rise of approximately 0.5 m is used for most engineering design in Ireland. However, wave setup on exposed coastlines and storm surges (low pressure and onshore winds) can significantly increase water levels.

Murphy (2014) argues that quantifying wave setup and storm surges would be better parameters to apply to engineering design along the Irish coastline.

In addition, Carbone et al. (2013) have studied wave impact on vertical cliffs and shown that certain wave groups may produce higher run ups than previously predicted (exceeding the initial wave amplitude by a factor of 5). These results have been backed up by other studies including Viotti et al. (2014) who showed that waves impinging on a vertical wall can exceed

six times the far field wave amplitude, while Herterich and Dias (2017) simulated the bathymetry of the Aran Islands and showed that run up can be amplified nearly twelve times under certain conditions. These studies suggest that the design wave heights used by engineers for coastal structures may be too low.

Tsunami surges result in coastal erosion, while tsunami-induced currents also present an obvious hazard to maritime activities and ports. An event like the 1755 Lisbon tsunami, which generated 2 m waves in Kinsale, nowadays would cause much

damage because of the larger number of boats and also pose a danger to human life. There is also the threat of major landslides off the west coast of Ireland (Salmanidou et al., in press) that could not only generate 4 to 5 m waves in Belmullet but also generate complex motions in the surrounding bays. While erosion can change the coastal landscape, so can accretion. More recently, thousands of tons of sand were deposited by unusual tides at Doonagh, Achill Island recreating the strand they had lost to storms 33 years prior (*The Irish Times* 1 May 2017).

## 8   Public Awareness and Education

### 8.1   Storm Waves and Rogue Waves

Most of the Irish coastline enjoys a seascape free from barriers or restrictions. This is one of the reasons that locals and foreigners alike are attracted to coastal amenities and stunning scenery. However, as evidenced from this catalogue there are dangers associated with extreme waves when venturing so close to the ocean. In particular, there are areas that seem to

be accident prone. Between Doonbeg and Kilkee, Co. Clare, 7 rogue or storm waves have caused accidents, while 5 such waves have been documented between Doolin and Fanore, Co. Clare. These sites may be more accident prone due to a higher frequency of extreme waves or, larger volumes of people doing coastal activities in the region. However, given that there have been multiple incidents where people have slipped and fallen into the sea in these areas, it is likely due to the volume of people.

Local knowledge may play a part in these incidents since there are many occurrences involving non-natives. It seems sensible

that signs should be present along certain parts of the coastline alerting people to the dangers of extreme waves. Signs do exist, for example see Figure **??**, however it may be more effective to emulate road signs identifying a "blackspot" region or, state the number of people who have died on Ireland's coastline. Warning signs should be combined with public education on the





subject. The Irish Coast Guard often run safety campaigns, for example in June 2016 they launched "No Life Jacket? No Excuse" aimed at people visiting Irish coastal areas over the summer holidays.

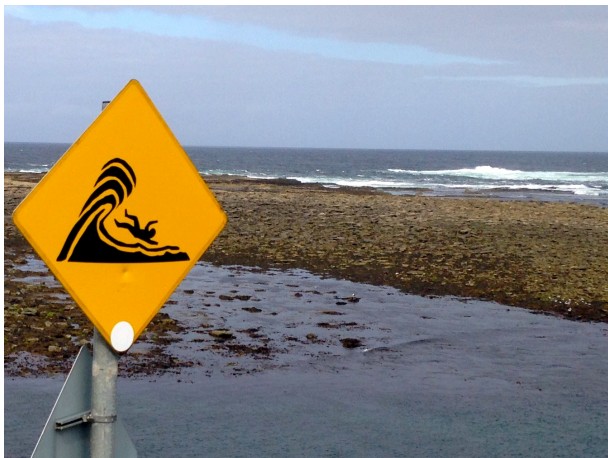

**Figure 19.** Warning sign at Kilkee, Co. Clare.

## 8.2    Storm Surges and Tsunamis

Storm surges and coastal flooding are somewhat predictable with modern day weather and tide forecasts. Warnings will often
be circulated by news and emergency services prior to a coastal flooding event. When flooding is expected in a region, safety measures are usually implemented by local county councils including the use of sandbags or advising vulnerable people to move to a safe place. The Office of Public Works also maintains a website to inform the public how to prepare for a flood and what to do during and after a flood [3]. However, the lead time, severity and location of flood forecasts could be improved. Funding for a new flood forecasting unit was announced by the Irish government in early 2016 and is due to be set up by Met Éireann and the Office of Public Works in 2017.

When it comes to tsunami forecasting, the International Tsunami Information Center provides warnings from the major tsunami warning centres around the world. So, if a tsunami were heading towards Ireland it is likely that a warning would be circulated. However, knowing that a tsunami is coming is not enough to avoid loss of life. Countries prone to tsunamis have pre-prepared risk maps and evacuation plans for different tsunami scenarios. This does not exist in Ireland due to the low risk of a life threatening tsunami occurring. However, there are two potential sources of large tsunamis that could effect Ireland. The first is a volcano landslide in the Canary Islands and the second, a large underwater landslide off the continental shelf in the Atlantic Ocean, but it is not clear what might trigger such an event.

On 27 September 2016, nations on the Atlantic, Mediterranean and Black Sea decided to strengthen their tsunami warning systems by giving France, Greece, Italy and Turkey a region-wide alert role and the UNESCO - Intergovernmental Oceano-

---

[3]*www.flooding.ie*




graphic Commission (IOC) NE-Atlantic and Mediterranean (NEAM) tsunami warning system is now fully operational. The Geological Survey of Ireland (GSI) is the lead organisation for tsunami emergency management in Ireland and the Irish delegate to the UNESCO-IOC NEAM tsunami warning group.

The GSI are developing tsunami warning and management systems with the assistance of The CENtre d'ALerte aux Tsunamis (CENALT), the tsunami Service Provider for the NE Atlantic. CENALT have included tsunami forecast points around the Irish coast so that tsunami arrival times can be provided within two minutes of having confirmation of seismic event parameters. Currently, Met Éireann provide a message receipt and forwarding service for tsunami warnings, but there is no Tsunami Warning Focal Point (TWFP) in Ireland to interpret and distribute messages if required. The GSI are moving forward to rectify this. The development of tsunami warnings and emergency management are important for Ireland and tsunami risk is
now on the official list of risks for Ireland. However, as discussed in 7.1 another aspect that needs to be explored are potential damages to the coast and coastal structures.

## 8.3   Other Dangers

Changing tides can also be a source of danger in Ireland. There are many incidents along Ireland's coastline where people have become trapped due to an incoming tide. For example in Galway Bay (*The Irish Times* 4 June 2012), in Bundoran, Co.
Donegal (*Afloat Magazine* 18 July 2014), at Green Island, Co. Clare (*The Irish Examiner* 22 March 2015) and in Bannow Bay Co. Wexford (*The Irish Independent* 21 November 2016). This is another circumstance where signs should be present in vulnerable areas, warning people to check the tide times.

Group swimming on Christmas day, St Stephens Day (26th December) or New years day have been occurring in towns across Ireland for many years, many are charity events while others are tradition. Those who take part need to be vigilant,
particularly as conditions can be dangerous at this time of year and there are not usually lifeguards on duty. On Christmas morning in 1997 at the Forty Foot, a famous swimming area in Sandycove, Co. Dublin, conditions were particularly bad and many of the enthusiastic, but non-regular swimmers got into trouble. One boy was swept away by a "sudden freak wave" and a teenage girl was "bounced at great speed from rock to wall to rock". (*The Irish Times*, 31 December 1997). Public education would encourage people to be safety conscious and make sensible decisions about whether to partake or not.

## 9   Services

Multiple organisations manage different aspects of the Irish coastline and marine sectors, some oversee safety while others manage data collection.

### 9.1   Data Services

The Office of Public Works (OPW) is a service organisation in Ireland, one of their main areas of responsibility is flood risk
management. They provide real time sea level data recorded at 11 tide gauges on their hydrometric network, including data up





to 5 weeks prior, through a web portal[4]. The OPW also maintain a website[5] that contains reports and information about floods that have occurred around Ireland.

The Marine Institute is the agency responsible for marine research, technology development and innovation in Ireland[6]. It provides access to realtime and past data from twenty tide level stations deployed around the Irish coast, along with 4 wave buoys and 5 weather buoys that are located in various locations offshore. They also provide a 6 day marine forecast for significant wave height, mean wave period and mean wave direction.

The Commissioners of Irish Lights (CIL) manages the Ireland's network of lighthouses, providing a safety and support service around the coast of Ireland. With advances of e-navigation the CIL now has a heavy focus on technology and data services. Real time wave and weather data can be retrieved for seven buoys located at various locations offshore and weather data at two lighthouses[7].

Met Éireann is the national weather service in Ireland. They provide real time and past weather observations along with weather forecasts and rainfall radar images. This includes sea area weather forecasts and sea state observations[8].

The OPW, in collaboration with the RPS Consulting Engineers and Met Éireann, are forecasting sea levels, tide and surge around the Irish coast. However this is not yet available to the public.

The Geological Survey of Ireland (GSI)[9] is responsible for acquiring geological data and providing advice and information in all aspects of Irish geology. In conjunction with the National Emergency Coordination Centre, the GSI will develop tsunami hazard maps and emergency response plans for Ireland. This will ensure the safety of coastal communities in the unlikely event of a tsunami hitting the Irish coast.

## 9.2 Rescue Services

The RNLI is a charity that was established in 1824 and is operated largely by volunteers. It provides lifeboat search and rescue services, seasonal lifeguards, water safety education and flood rescue response around the British Isles. There are 43 stations located around the island of Ireland.

The Irish Coast Guard is part of the Department of Transport, a department of the Irish government. Their overall objective is to reduce the loss of life on Ireland's seas, lakes, waterways and rivers, coastal and remote areas. Members are made up of both paid employees and unpaid volunteers. Unlike other countries, the Coast Guard in Ireland is not part of the defence forces. However they are assisted by the Air Corps and Navy.

The Community Rescue Boats Ireland are a group of independent voluntary rescue boats who make themselves available to the Irish Coast Guard, responding to emergencies in their area. Communities have traditionally set up teams following drowning tragedies.

---

[4]*waterlevel.ie/group/16*

[5]*www.floodmaps.ie*

[6]*www.marine.ie/Home/site-area/data-services/marine-data-centre*

[7]*www.irishlights.ie/technology-data-services/metocean-charts.aspx*

[8]*www.met.ie*

[9]*www.gsi.ie*



On the 12 September 2016 a member of the Doolin Coast Guard unit lost her life when the boat she was in capsized during a rescue operation. She was the first member of the Coast Guard to die on active duty.

## 10 Conclusions

This catalogue (in combination with O'Brien et al. (2013)) attempts to provide an extensive database of extreme waves around
the island of Ireland. 41 storm surges that caused severe coastal flooding, 12 tsunamis, 30 storm waves and 29 rogue waves are documented. This cannot be considered an exhaustive list, however it provides a benchmark to work from when considering how extreme waves impact Ireland. We hope that an accurate database can be established by combining this work with other sources. This is necessary in order to inform the future development of Ireland's marine resource and to protect the future of Irish coastlines and communities.

One of the major conclusions of this paper is that coastal flooding due to storm surges are common occurrences in Ireland with 68 events documented in this catalogue. Note that these are only events where flooding of more than half a metre was reported. There are numerous storm surges where flooding of more than half a metre is likely to have occurred but was not reported, or events where flooding was less than half a metre but still caused huge damage or disruption. This highlights the need for accurate documentation of events as they occur. Coastal flooding brings major socio-economic impacts, so it is
imperative that Ireland adequately prepares for such events now and into the future.

As found in O'Brien et al. (2013), rogue waves continue to be a common occurrence in Ireland with many people losing their lives as a result. Understanding the dynamics of these waves may reduce the associated risks in the future. However until then it is important that the public are educated on the subject and proper warnings are in place on dangerous stretches of coastline. Storm waves also pose major risks and this paper has shown that many of the very large waves recorded on the buoy network
are not rogue waves. Some of the largest waves recorded by Irish buoys are shown in Table 6, however the largest three should be treated with caution as readings can become unreliable beyond a certain threshold.

**Table 6.** Largest waves recorded in Irish marine territory

| Date | Wave Height | Location |
|---|---|---|
| 13th December 2011 | 20.4 m | M4 buoy |
| 12th February 2014 | 25 m | Kinsale Energy Gas Platform |
| 26 January 2014 | 23.4 m | M4 buoy |
| 26 January 2014 | 34 m | Killard wave rider buoy |
| 12th February 2014 | 31 m | Belmullet Berth A buoy |
| 2nd March 2016 | 36.4 m | Belmullet Berth B buoy |





With increasing average temperatures globally and rising sea levels, it is still uncertain how the marine climate will adjust in the future. This is an area of ongoing research and of particular interest to Ireland given its proximity to the ocean and the frequency of extreme events.

Finally, without the vital services and volunteers of the RNLI, Irish Coast Guard and Community Rescue Boats, the number

5   of deaths in Irish waters would be far higher. These services should continue to be well maintained and given proper recognition.

*Acknowledgements.* This work was funded by the ERC under the research project ERC-2011-AdG 290562-MULTIWAVE and ERC-2013-PoC 632198-WAVEMEASUREMENT. This study was also funded by Science Foundation Ireland (SFI) under the research project "Understanding Extreme Nearshore Wave Events through Studies of Coastal Boulder Transport" (14/US/E3111). The authors are grateful to ESBI for sharing the Killard wave measurements and would like to thank Teledyne R & D and Brian McConnell from GSI for their useful

10   contributions. Thanks also to Sarah Gallagher for her helpful comments on the paper. In addition, the authors thank Kathy Gordan from the Permanent Service for Mean Sea Level (PSMSL) for providing high frequency tide gauge recordings, Seamus Caulfield and Gretta Byrne for providing information on the damage caused in Co. Mayo during the 2014 storms. Thanks also to many people who passed on information and eye witness accounts of wave events, including David Long from the British Geological Survey, Paul Wignall from the University of Leeds, Sean Dineen from University College Dublin, Caroline from the Dunbeg Fort and Visitor Centre, Jim Hurley and Clive Hawkins.



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
