# Peer review of "Catalogue of extreme wave events in Ireland: revised and updated for 14680 BP - 2017"

_Natural Hazards and Earth System Sciences, 2017_

## Referee Comment (RC1) · E. Pelinovsky (Referee) · 28 Jul 2017

The reviewed paper extends and revises the previous catalogue of extreme wave events in Ireland published in 2013; it also adds the material for 2012-2016. The reviewer initiated the similar work with the rogue wave catalogue in the World Ocean, and now I am happy to see a very detailed catalogue for Ireland. I absolutely support the publication of the given paper.

Meanwhile, I have some minor comments:

1. Speaking about meteo-tsunami, it will be good to cite the reviewed paper: Vilibic I., Monserrat S., and Rabinovich A. Meteorological tsunamis on the US East Coast and in other regions of the World Ocean. Natural Hazards, 2014, vol. 74, 1-9.

2. It will be good to give a map with all locations of extreme wave observations.

[Figure]

3. I am not sure that the title properly reflects the context. Perhaps, the title 'Catalogue of extreme wave events in Ireland: revised and updated for 14680 BP – 2016" will be better.

4. Figure 2 and 3. It will be interesting to indicate by an arrow the tsunami entry time or, if it is difficult the earthquake time.

5. Section 5.2 discusses the rogue wave events, but I could not find the magnitude of the significant wave heights in any of the section items. As a result, it is difficult to check the criterion for rogue waves and to judge its validity. Such criterion, for instance, was specially checked in the rogue wave catalogue by I. Nikolkina and I. Didenkulova.

6. Perhaps, the citing of the paper "Baschek, B., and J. Imai. Rogue wave observations off the US West Coast. Oceanography, 2011, vol.24, 158–165, doi:10.5670/oceanog.2011.35" will be useful to give more information about the existing rogue wave catalogues.

I recommend publishing this paper after some minor revision.

---

## Editor Comment (EC1) · M. Gonzalez (Editor) · 31 Aug 2017

Dear authors,

Please, includes in your response to the reviewers (as an attachment) the corrected manuscript including in it all your corrections and modifications.

Mauricio G. NHESS Editor

---

## Short Comment (SC1) · 12 Sep 2017

The purpose of the paper is clear, to provide a survey of extreme wave events in Ireland - in the form of a catalogue - extending the existing catalogue by O'Brien (2013).

The documentation is important and unique.

Some comments:

On p. 2, section 2.2.1, Tsunamis, it is now documented that large ships moving over depth changes are a new source of appreciable tsunamis (mini-tsunamis) contributing to a new danger and a new source of erosion in coastal waters, as documented in J. Grue. Ship generated mini-tsunamis, J. Fluid Mech. Vol. 816, pp. 142-166. This should be referred to in section 2.2.1.

Other comments to the text: The texts in the figures are sometimes difficult to read

(the text is very small). In some figures the scales / axes are not given, so they are in fact useless. An example is figure 13 on p- 29. Figure 11 is very small. Figure 6 is interesting but is useless without the vertical axis (the zero-level can be estimated).

Another point: It would increase the value of the data to provided both the wave height and the crest height, where the latter is a useful parameter in judging the local wave slope (when the period is known). E.g. section 3.1.13.

John Grue, University of Oslo, Norway

---

## Short Comment (SC2) · 1 Oct 2017

Data from floating buoys should be taken with a grain of salt: attached is an example of buoy records in 20 m depth off "Truc Vert" beach (see context in paper by Senechal et al. 2011 DOI: 10.1007/s10236-011-0472-x).

Indeed, buoys only measure accelerations, and a double integration provides displacement. Also the buoy sensor in the case of a usual Datawell is gimballed in the buoy and can produce spurious large values if the buoy is rotated. I would suggest that the author go back to the raw data archived in the datalogger and not the data transmitted via HF, it should not have the clipping at +/- 20 m.

Also a spectral analysis of the record can be used to check for unrealistic long periods at the time of the extreme waves. The pattern of waves plotted in figure 11 and 12 look suspicious. What is the depth of the buoy? How was the data retrieved? If obtained

[Figure]

via HF transmission, was the data QC flag at 0?

[Figure]

[Figure]

Figure2: Another example of bad signal in the 2008/03/11 01:00 record. Note that the vertical scale is in meters. Top panel is the HF transmission record and bottom panel is the datalogger record. It is rather surprising that the HF link functioned properly in these conditions.

**Fig. 1.** Example of Datawell bad data

---

## Short Comment (SC3) · 1 Oct 2017

The directional properties (a1,b1,a2,b2) can also be used to estimate a directional wave spectrum, which usually becomes very broad when there is bad data...

---

## Referee Comment (RC2) · Anonymous Referee #2 · 5 Oct 2017

The comment was uploaded in the form of a supplement:
https://www.nat-hazards-earth-syst-sci-discuss.net/nhess-2017-206/nhess-2017-206-RC2-supplement.pdf

---

## Author Comment (AC1) · 22 Oct 2017

**Referee #1 comments: E. Pelinovsky**

1. Speaking about meteo-tsunami, it will be good to cite the reviewed paper: Vilibic I., Monserrat S., and Rabinovich A. Meteorological tsunamis on the US East Coast and in other regions of the World Ocean. Natural Hazards, 2014, vol. 74, 1-9.
   *A reference will be added to Section 2.21.*

2. It will be good to give a map with all locations of extreme wave observations.
   *Two maps will be added showing the short and long wave events.*

3. I am not sure that the title properly reflects the context. Perhaps, the title 'Catalogue of extreme wave events in Ireland: revised and updated for 14680 BP – 2016" will be better.
   *We plan on changing the title to something along these lines.*

4. Figure 2 and 3. It will be interesting to indicate by an arrow the tsunami entry time or, if it is difficult the earthquake time.
   *We will add boxes around the visible oscillations in the tide gauge figures.*

5. Section 5.2 discusses the rogue wave events, but I could not find the magnitude of the significant wave heights in any of the section items. As a result, it is difficult to check the criterion for rogue waves and to judge its validity. Such criterion, for instance, was specially checked in the rogue wave catalogue by I. Nikolkina and I. Didenkulova.
   *We will add significant wave height information where it is available. However, some incidents that have been categorized as rogue waves are often associated with accidents and no physical recordings exists.*

6. Perhaps, the citing of the paper "Baschek, B., and J. Imai. Rogue wave observations off the US West Coast. Oceanography, 2011, vol.24, 158–165, doi:10.5670/oceanog.2011.35" will be useful to give more information about the existing rogue wave catalogues.
   *A paragraph will be added to Section 2.12 based on this paper.*

**Referee #2 comments: Anonymous**

This paper is an updated catalogue on extreme sea waves that occurred in the Irish waters. These extreme wave events are separated into two classes: short waves (rogue and storm waves) and long waves (tsunami waves and storm surges). With this classification the Killard wave belongs to both categories because this rogue wave occured in shallow water (kh = 0.74).

***It is true that the Killard wave is an intermediate wave bordering on the boundary between short and long waves, however we feel that it should be classified as a short (rogue) wave. Referring to Section 2 of the paper, kh ranges are provided for short (1 - $10^3$) and long waves ($10^{-3}$ - $10^{-1}$), the kh value for the Killard wave kh = 0.74 is closer the to range given for short waves than for long waves.***

Many extreme sea wave events are carefully reported and conclusions are given to protect communities and the Irish coastlines and suggestions for future researches in Ireland marine resources are provided.

Page 36, two papers are cited, one claims that extreme sea wave events may increase whereas the other claims the opposite. May the authors give more convincing argument to justify this disagreement?

***The first of the 2 papers cited (Gleeson et al., 2013) is based on projections with coarse resolution, while the second paper (Gallagher et al., 2016) is based on a much higher resolution. Therefore, we believe the second paper provides a more accurate representation of the future wave climate. We will remove the reference to the first paper in the context of wave climate projections.***

About rogue wave events, I find that an explanation attempt on possible physical mechanisms generating these huge wave is missing.

***There is a discussion about the physical mechanisms of rogue waves given in Section 2.1.2 (page 4 in particular).***

To conclude, this is an interesting paper that could serve as an example for other countries threatened by this abnormal events. I support the publication of this paper in NHESS.

**Short Comment #1: John Grue**

The purpose of the paper is clear, to provide a survey of extreme wave events in Ireland - in the form of a catalogue - extending the existing catalogue by O'Brien (2013). The documentation is important and unique.

Some comments:
On p. 2, section 2.2.1, Tsunamis, it is now documented that large ships moving over depth changes are a new source of appreciable tsunamis (mini-tsunamis) contributing to a new danger and a new source of erosion in coastal waters, as documented in J. Grue. Ship generated mini-tsunamis, J. Fluid Mech. Vol. 816, pp. 142-166. This should be referred to in section 2.2.1.
***This category of tsunami and the above reference will be added to Section 2.2.1.***

Other comments to the text: The texts in the figures are sometimes difficult to read (the text is very small). In some figures the scales / axes are not given, so they are in fact useless. An example is figure 13 on p-29. Figure 11 is very small. Figure 6 is interesting but is useless without the vertical axis (the zero-level can be estimated).
***The size of the fonts on the axes will be increased in figures where the text is difficult to read (eg. Figures 11, 13). While Figure 6 doesn't provide much information without a proper vertical axis, the text on the top right ("Feet 25") and bottom right ("25") of the figure give an indication of the scale. We believe this figure is an interesting one and since an explanation of the measurement is provided in the text (Section 4.2.2: "a wave that is 4.1 times the average wave height") it is worthy of keeping this figure in the paper. We will add an explanation of the measurement to the figure caption as a way to improve understanding the figure.***

Another point: It would increase the value of the data to provided both the wave height and the crest height, where the latter is a useful parameter in judging the local wave slope (when the period is known). E.g. section 3.1.13.
***Wave height and crest height will be added where it is possible to retrieve this data.***

**Short Comments #2 and #3: F. Ardhuin**

**2**

Data from floating buoys should be taken with a grain of salt: attached is an example of buoy records in 20 m depth off "Truc Vert" beach (see context in paper by Senechal et al. 2011 DOI: 10.1007/s10236-011-0472-x).
*We fully agree with the reviewer and in fact there are several warnings in the text about measurements from floating buoys.*

Indeed, buoys only measure accelerations, and a double integration provides displacement. Also the buoy sensor in the case of a usual Datawell is gimballed in the buoy and can produce spurious large values if the buoy is rotated. I would suggest that the author go back to the raw data archived in the datalogger and not the data transmitted via HF, it should not have the clipping at +/- 20 m.
*Whenever we could, we went back to the original datalogger readings. In some case, we do not have access to them. One example is the Kinsale Energy Gas Platform, which recorded a maximum wave height of 25 m on the 12th February 2014 during a severe windstorm (Met Éireann, 2016a). We indirectly checked the raw data and the measurement was confirmed.*

Also a spectral analysis of the record can be used to check for unrealistic long periods at the time of the extreme waves. The pattern of waves plotted in figure 11 and 12 look suspicious. What is the depth of the buoy? How was the data retrieved? If obtained via HF transmission, was the data QC flag at 0?
*The depth of the buoy is 39m and we had access to the raw data. We were conservative with the QC fags and removed anything that wasn't 0 (or whatever the code was for "fine"). These 2014 wave buoy measurements were the trigger for us to put an ADCP at the same location the next winter. We also measured 20m waves with the ADCP. We also wanted to compare wave buoy measurements with ADCP measurements, but the wave buoy broke down just before the ADCP was deployed.*

**3**

The directional properties (a1,b1,a2,b2) can also be used to estimate a directional wave spectrum, which usually becomes very broad when there is bad data...
***Thank you. We will make sure to use the directional properties when estimating the directional wave spectra in future papers.  But it is beyond the scope of the present catalogue.***

---

## Author Comment (AC2) · 22 Oct 2017

The comment was uploaded in the form of a supplement:
https://www.nat-hazards-earth-syst-sci-discuss.net/nhess-2017-206/nhess-2017-206-AC2-supplement.pdf

---

## Author Comment (AC3) · 22 Oct 2017

The comment was uploaded in the form of a supplement:
https://www.nat-hazards-earth-syst-sci-discuss.net/nhess-2017-206/nhess-2017-206-AC3-supplement.pdf

---

## Author Comment (AC4) · 22 Oct 2017

The comment was uploaded in the form of a supplement:
https://www.nat-hazards-earth-syst-sci-discuss.net/nhess-2017-206/nhess-2017-206-AC4-supplement.pdf
* * *

---

## Author Comment (AC6) · 22 Oct 2017

The comment was uploaded in the form of a supplement:
https://www.nat-hazards-earth-syst-sci-discuss.net/nhess-2017-206/nhess-2017-206-AC6-supplement.pdf

---

## Referee Report (RR1)

Report on the paper "Catalogue of extreme wave events in Ireland: revised and updated for 14680 BP-2017" by L. O'Brien *et al.*

About the classification of Killard rogue wave:

(i) Fenton (JFM, 94(1), 1979) demonstrated exactly that the boundary of applicability between short waves (deep) and long waves (shallow) is at  $\lambda/d \approx 8$  which corresponds to  $kd \approx \pi/4$

(ii) From an empirical point of view there are several relations to define this boundary. In the book of R.G. Dean & R.A. Dalrymple it  $kd \approx \pi/10$  and in the book of M.W. Dingemens it is  $kd \approx 2\pi/10$ . I found in a paper by Fenton this relation  $H\lambda^2/d^3 > 40 \Rightarrow \log$  wave.

It is clear that the empirical definitions classify the Killard wave as a short wave and on the opposite the more rigorous definition classifies it as a long wave. Consequently, I suggest to the authors to discuss this point in their revised version.

---

## Author Response (AR2)

**Reviewer #2 Comments:**

About the classification of Killard rogue wave:

(i) Fenton (JFM, 94(1), 1979) demonstrated exactly that the boundary of applicability between short waves (deep) and long waves (shallow) is at $\lambda/d \approx 8$ which corresponds to $kd \approx \pi/4$

(ii) From an empirical point of view there are several relations to define this boundary. In the book of R.G. Dean & R.A. Dalrymple it $kd \approx \pi/10$ and in the book of M.W. Dingemens it is $kd \approx 2\pi/10$. I found in a paper by Fenton this relation $H\lambda^2/d^3 > 40 \Rightarrow$ long wave.

It is clear that the empirical definitions classify the Killard wave as a short wave and on the opposite the more rigorous definition classifies it as a long wave. Consequently, I suggest to the authors to discuss this point in their revised version.

**Author's Response:**

We thank both reviewers for their comments and helpful information.

In response to the above comments, we have added a paragraph to Section 2 (page 2) discussing the different classifications that exist between long and short waves.

In addition, we have added a link to an online interactive map that allows users to navigate their way through events discussed in this paper. This appears in section 10 and included as supplementary material also.

[revised manuscript text omitted]